# Color image encryption algorithm based on ∞-shaped transformation and closed-loop control model

**Feng Zhao[1], Xiaoqiang Zhang[2]\*, Fang Zhu[1]**

**1** Department of General Education, Anhui Xinhua University, Hefei, Anhui, China, **2** School of Information and Control Engineering, China University of Mining and Technology, Xuzhou, Jiangsu, China

\* grayqiang@163.com

## Abstract

As color images have become a cornerstone of information exchange in the digital age, ensuring their security is of paramount importance. With the traditional scrambling–diffusion structure, this paper proposes a novel color image encryption algorithm by integrating of an ∞-shaped transformation with a closed-loop control mechanism. First, the three channel matrices are merged, and the elements in each row are linked into a closed loop for initial diffusion. Secend, the diffused image is subsequently restructured into a three-row matrix and scrambled using a unique ∞-shaped transformation. Finally, column-wise closed-loop diffusion is applied to generate the cipher image. This algorithm not only achieves effective inter-channel pixel confusion through the ∞-shaped transformation, but also performs additional diffusion and confusion under the closed-loop control model. Experimental results demonstrate the algorithm's excellent overall performance: the key space is as large as $2^{413}$, information entropy approaches the ideal value of 8 with increasing image size, and the algorithm exhibits high sensitivity, with NPCR and UACI exceeding 99.6% and 33.4%, respectively. Quantitative evaluation confirms that the proposed algorithm offers strong robustness against differential, statistical, and brute-force attacks.

## 1. Introduction

In the era of information explosion, images have become a vital medium for information storage and transmission due to their rich visual expressiveness and characteristics such as high information density, intuitive representation, and ease of dissemination. They are widely applied in many domains including daily life, industrial production, healthcare, and finance. However, the increasing frequency of image transmission over networks has led to a rise in incidents of image theft, tampering, and damage, thus positioning image security as a core concern in information security research. Digital image encryption offers a direct and effective

**Data availability statement:** All relevant data are within the manuscript.

**Funding:** The work is supported partly by the Key Research Project of Natural Science in Universities of Anhui Province under Grant No. 2023AH051807, Grant No.2022AH051864, and Grant No. 2024AH050611, and Anhui Xinhua University Level Research Project under Grant No.2022zr003.

**Competing interests:** NO authors have competing interests Enter: The authors have declared that no competing interests exist.

solution to this challenge. By altering pixel positions and values, it transforms information-rich images into random noise-like patterns, making it extremely difficult to extract any meaningful content. With the rapid advancement of communication and computer technologies, significant breakthroughs have been made in image encryption technology, giving rise to innovative methods grounded in diverse theories and techniques such as chaos theory [1–6], cellular automata [7], DNA coding [8–10], neural networks [11,12], and compressed sensing [13,14].

Chaotic systems are characterized by extreme sensitivity to initial conditions and parameters, inherent ergodicity, and high pseudorandomness, which naturally align with the core cryptographic requirements of confusion and diffusion, rendering them a cornerstone technology in the field of image encryption. Such systems can be broadly categorized into continuous and discrete chaotic maps. Since the Lorenz system was introduced as the first dissipative model numerically verified to exhibit chaotic behavior, Meng *et al.* [15] significantly extended its effective parameter range by incorporating a nonlinear regulation mechanism, thereby substantially expanding the key space. Sambas *et al.* [16,17] constructed a new three-dimensional chaotic system featuring a peanut-shaped closed curve of equilibrium points and a four-dimensional hyperjerk chaotic system with a half-line equilibrium, both of which were employed to design efficient image encryption algorithms. Xu *et al.* [18] constructed a four-dimensional hyperchaotic system and developed a corresponding encryption strategy integrated with an Arnold perturbation module. Benkouider *et al.* [19] further extended this framework to higher-dimensional hyperchaotic systems. However, despite their strong theoretical performance, the practical deployment of continuous chaotic systems remains constrained by several limitations, including finite precision effects, high computational overhead, complex parameter configuration, cumbersome key management, and vulnerability to side-channel attacks. These issues collectively hinder the efficiency of random sequence generation in real-time encryption scenarios. Consequently, research focus has progressively shifted toward discrete chaotic maps. Notably, the Exponential–Sine–Cosine (ESC) map proposed by Kumar and Dua [20] effectively overcomes the structural bottlenecks of one-dimensional discrete maps in terms of key space and complexity, and has been successfully embedded into the scrambling and diffusion stages of image encryption, achieving robust security performance. Du *et al.* [21] designed a discrete hyperchaotic system with a hybrid cross-feedback mechanism, integrating the two-dimensional Logistic map, the one-dimensional Cubic map, and the Feigenbaum map. Experimental results demonstrate that the system exhibits superior ergodicity, complex nonlinear dynamical behavior, and a broad hyperchaotic parameter range, significantly enhancing its cryptographic suitability. To address the dynamical degradation commonly observed in discrete chaotic maps under finite precision, Wang *et al.* [22] proposed an exponential chaotic system with time-varying delay control and developed a visual encryption strategy tailored to sensitive image regions. Experimental validation confirms the method's excellent performance in terms of information entropy, resistance to differential attacks, and pixel correlation decorrelation. Zhao *et al.* [23] introduced a two-dimensional dual-absolute-value logistic–fractional chaotic map, which integrates

the dynamical characteristics of logistic, fractional-order, sine, and cubic maps. Combined with a DNA coding mechanism, this approach forms an efficient multi-image encryption framework.

Image encryption predominantly involves permutation and diffusion mechanisms. Demirtaş [24] introduced a U-shaped scanning method for positional scrambling, inversely rearranging rectangular pixel blocks. Wen *et al*. [25] applied integer wavelet transforms to transition the image from the spatial domain to the frequency domains, encrypting information-rich low-frequency data using bit permutation, three-dimensional (3D) S-box substitution, ciphertext interleaving, and diffusion. Zhang *et al*. [26] detailed a multi-step process encompassing zigzag spiral scrambling, cross bit-plane transformation, and DNA encoding-decoding for final encryption. Chaotic sequences guided pixel stacking in [27], coupled with sliding window-based block iffusion. Image padding facilitated uniform segmentation in [28], preceding inter-block scrambling, intra-block Josephus permutations, and concluding with padding removal and comprehensive zigzag scrambling. Zhang *et al*. [29] initially employ a pseudo-wavelet transform to decompose the image. This is followed by generating pseudo-random sequences using a 2D chaotic map for pixel-level processing of low-frequency components and block-level processing of detail components. Gao *et al*. [30] utilize a newly constructed chaotic system to design a 3D coordinate matrix for multi-image scrambling operations, subsequently performing pixel diffusion through XOR operations to obtain encrypted images. Chowdhury *et al*. [31] proposed a thumbnail-preserving encryption scheme based on an improved two-dimensional piecewise logistic map, which achieves lossless decryption while preserving the visual content of thumbnails and increases the encryption speed by approximately 17 times.Through the persistent efforts of researchers, image encryption algorithms have emerged in an endless stream.

However, the advancement of cryptanalysis techniques has kept pace, continually testing and revealing vulnerabilities in existing schemes. Recent cryptanalytic studies have successfully exposed security flaws in several notable encryption frameworks. For instance, analyses have been conducted on schemes based on binary bit-plane extraction and multiple chaotic maps [32], color image encryption using fractional-order chaos [33], and methods incorporating the Feistel network with dynamic DNA encoding [34]. These works highlight common pitfalls such as inadequate randomness in chaotic sequences, insufficient diffusion properties, or structural weaknesses that can be exploited by chosen-plaintext or known-plaintext attacks. Therefore, it is of critical importance for any newly proposed encryption algorithm to not only introduce novel mechanisms but also to rigorously demonstrate its resilience against these established and evolving cryptanalytic methodologies. Motivated by this imperative, our work proposes a novel color image encryption scheme. The main contributions of this paper are described as follows.

[1]  A novel 2D-CSHS is proposed. Through an analysis of its phase diagrams, Lyapunov exponents, and randomness characteristics, we demonstrate that this chaotic map exhibits excellent nonlinear dynamical properties.

[2]  An innovative $\infty$-shaped transformation method is developed, which effectively scrambles elements within matrices. When applied to image encryption systems, it successfully disrupts pixel information across all three channels of color images.

[3]  A closed-loop control model is established, where pixel vectors are refactored into closed loops with randomly selected starting positions for diffusion. This algorithm simultaneously achieves both position scrambling and value diffusion through closed-loop operations.

[4]  A novel color image encryption algorithm is designed by integrating the $\infty$-shaped transformation and closed-loop control model. Experimental results confirm that the proposed algorithm features sufficiently large key space, high key sensitivity, and strong robustness against attacks.

The remainder of this paper is structured as follows: Section 2 introduces a novel chaotic map and validates its randomness properties. Section 3 outlines the foundational principles of the U-shaped transformation and the closed-loop control model. Section 4 elaborates on the proposed encryption algorithm, detailing operational steps and providing

corresponding pseudocode. Section 5 discusses simulation results and evaluates the performance of the proposed scheme. Section 6 concludes the paper.

## 2. 2D-CSHS

This section formally defines a two-dimensional chaotic system and systematically evaluates its chaotic performance using well-established metrics, including bifurcation diagrams, Lyapunov exponents (LEs), sample entropy (SE), and the NIST SP 800−22 statistical test suite. To further substantiate the superiority of the proposed 2D-CSHS, a comparative analysis is conducted against four representative two-dimensional chaotic maps under identical experimental conditions.

### 2.1. Construction of 2D-CSHS

The one-dimensional Logistic map is a relatively classic chaotic map. Based on this, Ref. [35] proposed an improved cubic map, whose mathematical equation is given by Eq. (1):

$$x_{n+1} = u\left(x_n - x_n^3\right),$$

(1)

where $x_n \in (0,1)$ represents the state variable. u is the control parameter, which exhibits chaotic behavior within the range (2.5, 3).

The sine function is also a type of nonlinear function [36], and its mathematical expression is given by Eq. (2):

$$x_{n+1} = \sin\left(\frac{a}{x_n}\right),$$

(2)

where $a \in (0,+\infty)$ is the control parameter.

To enhance chaotic performance and construct diverse chaotic systems, this paper proposes a novel 2D- CSHS based on the composition of trigonometric and nonlinear functions by combining cosine and sine functions. Its mathematical representation is given by Eq. (3):

$$\begin{cases} x_n = \cos\left(u\pi\left(x_{n-1} - y_{n-1}^3\right)\right) \\ y_n = \sin\left(r\left(\frac{\pi^2}{x_{n-1}}\right) + y_{n-1}\right) \end{cases},$$

(3)

where u and r are control parameters with $u \in [1,10]$ and $r \in [5,10]$. According to Eq. (3), $x_n \in [-1,1]$ and $y_n \in [-1,1]$. As a result, singularities may occur during the iterative process when $x_{n-1} = 0$. In practical numerical iterations, a regularization method is adopted to address this issue: when $|x_{n-1}| < \varepsilon$, its value is adjusted to $x_{n-1} \leftarrow x_{n-1} + \text{sgn}\left(x_{n-1}\right) \cdot \varepsilon'$, where $\varepsilon'$ is set to $10^{-10}$. This approach effectively avoids divergence caused by singularities. Since only a negligible perturbation is introduced, the overall chaotic dynamics of the system remain largely unchanged.

Compared with their one-dimensional counterparts, high-dimensional hyperchaotic maps generally exhibit more complex dynamical behaviors. In recent years, coupling, cascading, and combinatorial strategies have been employed to construct two-dimensional chaotic maps, as summarized in Table 1.

### 2.2. Bifurcation diagram

The bifurcation diagram provides a visual representation of the dynamical behavior of a chaotic system as parameters change, aiding in the identification of chaotic characteristics. Fig 1 shows the bifurcation diagrams of x and y under parameter variations.

**Table 1. Four proposed 2D chaotic maps.**

| Reference | Name | Definition | Control parameters. |
|---|---|---|---|
| [37] | 2D-SCMLCI | $\begin{cases} x_i = r\sin\left(\pi\left((y_{i-1}+h)k\sin(a\pi/x_{i-1})\right)\right) \\ y_i = r\sin\left(\pi\left((kx_i+h)\sin(a\pi/x_{i-1})\right)\right) \end{cases}$ | $r, h, k, a$ |
| [38] | 2D-LSM | $\begin{cases} x_i = \cos(4ax_{i-1}(1-x_i)+b\sin(\pi y_i)+1) \\ y_i = \cos(4ay_{i-1}(1-y_{i-1})+b\sin(\pi x_{i-1})+1) \end{cases}$ | $a, b$ |
| [39] | 2D-LCCCM | $\begin{cases} x_i = \cos\left(\pi^2\left(4\mu x_{i-1}\left(1-x_{i-1}\right)+py_{i-1}\left(1-y_{i-1}^2\right)\right)+\pi/2\right) \\ y_i = \cos\left(\pi^2\left(4\mu y_{i-1}\right)\left(1-y_{i-1}\right)+px_i\left(1-x_i^2\right)+\pi/2\right) \end{cases}$ | $u, p$ |
| [40] | 2D-ELMM | $\begin{cases} x_i = e^a x_{i-1}(e^b y_{i-1}-1) \qquad \mathrm{mod}\ 1 \\ y_i = e^b y_{i-1}(e^a x_{i-1}-1) \qquad \mathrm{mod}\ 1 \end{cases}$ | $a, b$ |

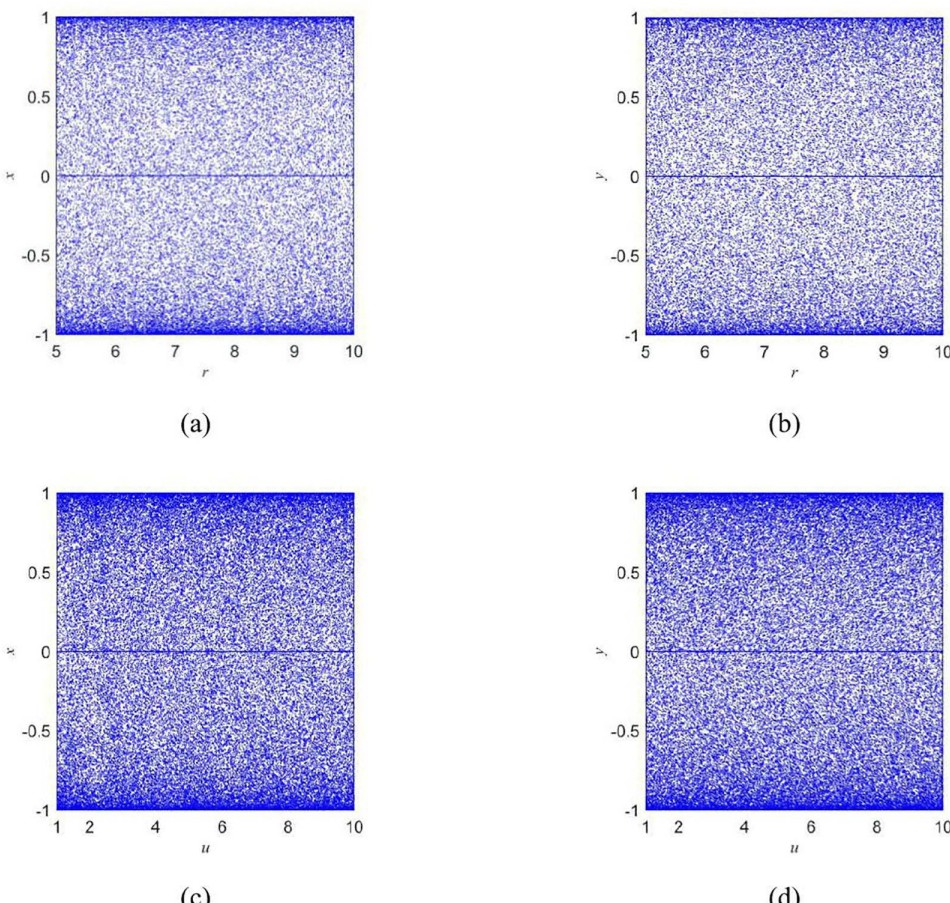

**Fig 1. Bifurcation diagrams.** (a) Bifurcation diagram of the $x$ with fixed $u=1$ varying $r$; (b) Bifurcation diagram of the $y$ with fixed $u=1$ varying $r$; (c) Bifurcation diagram of the $x$ with fixed $r=5$ varying $u$; (d) Bifurcation diagram of the **$y$** with fixed r=5 varying **u.**

The bifurcation diagram offers a visual depiction of the dynamical behavior of a chaotic system as parameters vary, facilitating the identification of chaotic traits. Fig 1 illustrates the bifurcation diagrams for $x$ and $y$ with fixed parameters ($u=1$, $r=5$). From the figure, the 2D-CSHS exhibits no periodic windows and has a uniform value distribution across the parameter ranges of $1 \leq u \leq 10$ and $5 \leq r \leq 10$.

### 2.3. Phase diagram

The phase portrait offers a clear and intuitive depiction of the evolutionary trajectory of a chaotic system under specified conditions. Fig 2 presents the phase portrait of the proposed chaotic map, generated with initial conditions $(x_0, y_0) = (0.3, 0.5)$ and control parameters $u=1$ and $r=5$. The resulting phase portrait indicates that the system's outputs are distributed across the entire phase plane, confirming the chaotic system's strong random output characteristics and ergodicity.

### 2.4. Lyapunov exponent

LE is a critical measure for assessing the dynamics of chaotic systems, encapsulating their complexity and sensitivity to initial conditions. It is defined as follows:

$$LE = \lim_{N \to \infty} \frac{1}{N} \sum_{i=1}^{N} \ln \left| f'(x_i) \right|.$$

(4)

If the LE is greater than 0, the system exhibits chaotic behavior. Furthermore, when two or more LEs are positive, the randomness and complexity of the system are enhanced. Fig 3 shows the variations of the Lyapunov exponents of the proposed map with respect to its two parameters. It can be observed that when $u=3.5$ is fixed and $r$ varies within [5, 10], both LEs remain positive. Similarly, when $r=5$ is fixed and $u \in [1, 10]$, both LEs exceed 2 and show a steady increase.

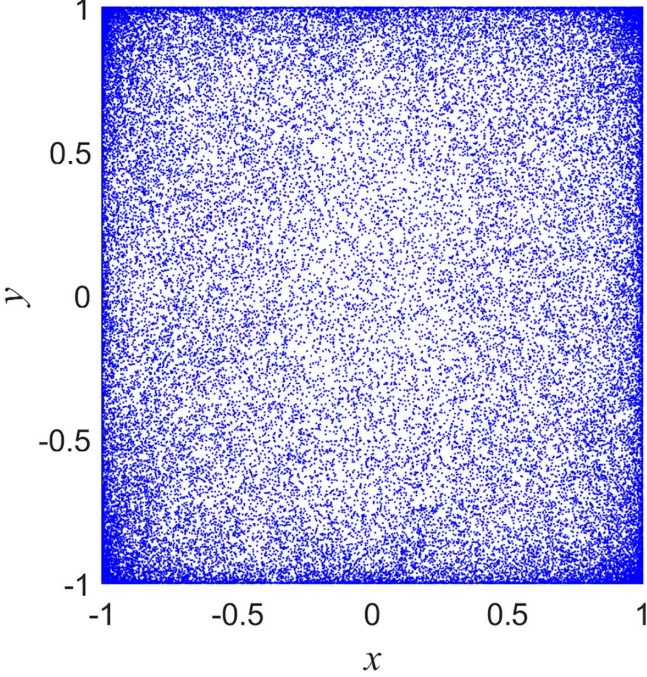

**Fig 2. Phase diagrams of the proposed 2D-CSHS.**

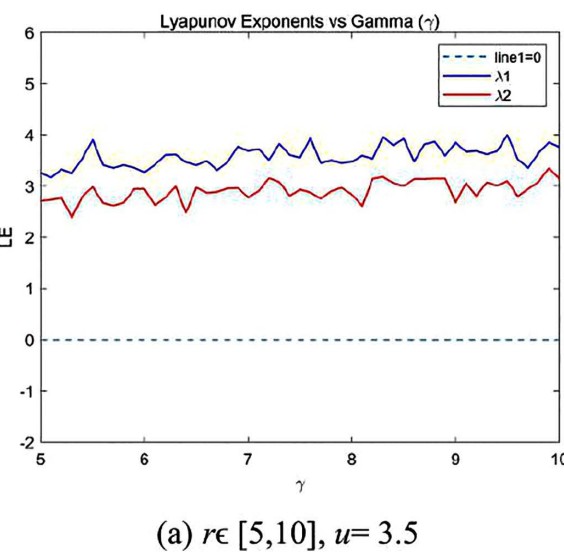

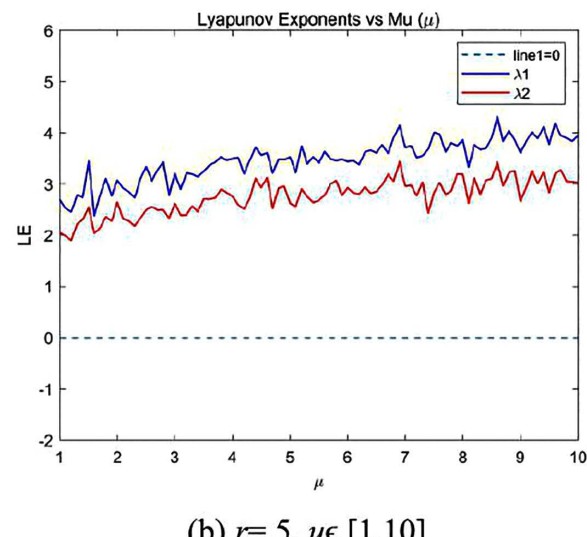

(a) $r \in [5,10]$, $u = 3.5$ (b) $r = 5$, $u \in [1,10]$

**Fig 3. LEs of the proposed 2D-CSHS.**

Fig 4 presents the heat map of the largest Lyapunov exponent (LLE). The figure indicates that for $r \in [5, 10]$ and $u \in [1, 10]$, the LLE remains above 2.8. To ensure encryption security, we selected parameter ranges within these intervals to guarantee pronounced chaotic behavior in the system.

Based on a comprehensive consideration, the parameter intervals were set to $u \in [1, 10]$ and $r \in [5, 10]$. In addition, comparative experiments were conducted in this study, in which the parameters of each map were normalized. As shown in Fig 5, the 2D-CSHS exhibits a continuous chaotic parameter range, and its LLE is significantly higher than those of the other maps under comparison. In other words, the 2D-CSHS outperforms the other maps in terms of the Lyapunov exponent.

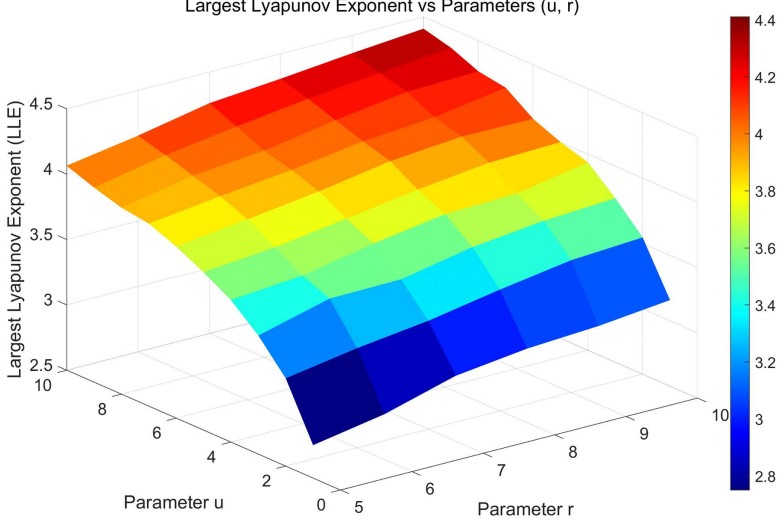

**Fig 4. Heatmap of the LLE as parameters (u,r) vary.**

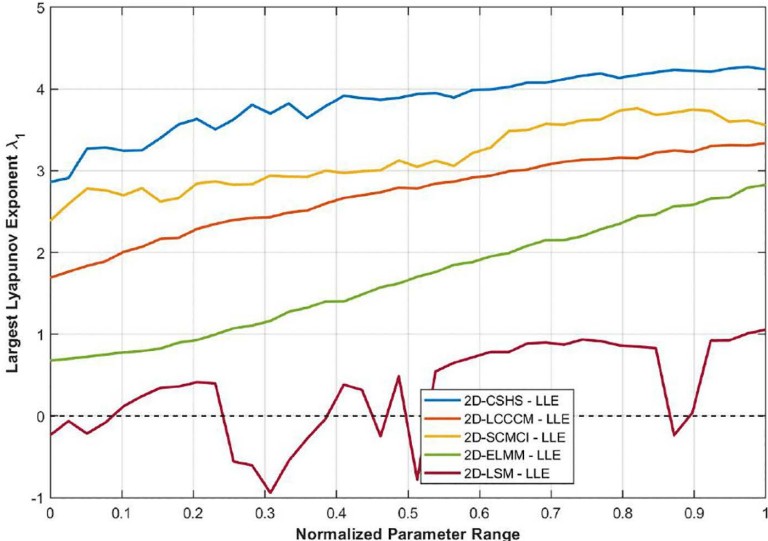

**Fig 5. LLE comparison between the proposed 2D-CSHS and four other 2D chaotic maps.**

## 2.5. Sample entropy

Sample entropy (SE) is a quantifiable metric that assesses the complexity and irregularity of time series data, with higher values reflecting increased intricacy and lower ones indicating more regular patterns. The oscillatory behavior within these series, as captured by SE, demonstrates a direct relationship with the sequence's complexity level, which can be calculated through the procedure outlined in Eq. (5) [41].

$$S(m, r, N) = -\log(\frac{S}{P}),$$

$$(5)$$

where $m$, $r$ and $N$ represent the template vector size, tolerance, and time series length, respectively, while S and P denote the Chebyshev distances between template vectors at positions $i$ and $j$. As shown in Fig 6, with parameters ranging from 5 to 10, the sample entropy values consistently exceed 1.5. These results demonstrate that the proposed 2D-CSHS exhibits strong complexity and irregularity characteristics.

## 2.6. Sensitivity analysis

Sensitivity analysis provides a methodological framework for evaluating the degree of responsiveness of a chaotic system's output to variations in input parameters. Introducing minute perturbations (on the order of $10^{-15}$) to initial values results in divergent pseudorandom sequences, serving as a definitive indicator of high sensitivity to initial conditions. Fig 7 illustrates that infinitesimal increments ($10^{-15}$) in either the $x$ or $y$ initial values lead to significant divergence in both $x$ and $y$ sequences. This empirical evidence conclusively demonstrates that the 2D-CSHS exhibits exceptional sensitivity to initial conditions.

## 2.7. NIST SP 800−22 statistical test

The NIST SP 800−22 test, developed by the National Institute of Standards and Technology (NIST), provides a comprehensive methodology for evaluating the randomness of binary sequences generated by pseudorandom number generators (PRNGs). This test suite consists of 15 distinct statistical tests, each designed to detect specific patterns

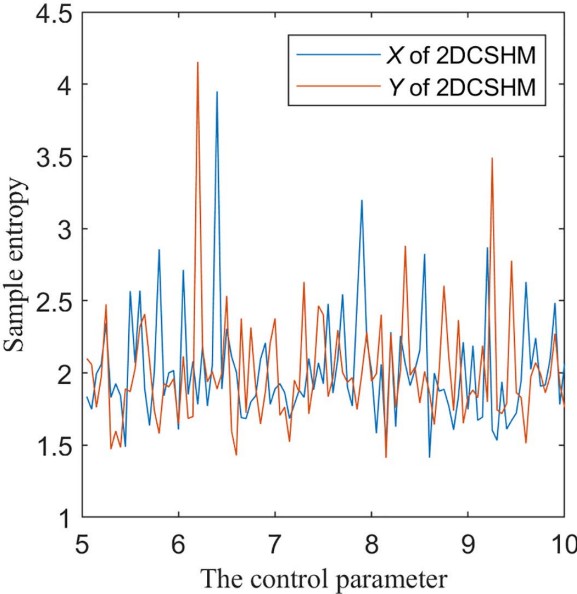

**Fig 6. Sample entropy of the proposed 2D-CSHS.**

of non-randomness in sequences. Notably, several tests are decomposed into multiple sub-tests, with each evaluation producing an associated p-value. According to NIST standards, sequences demonstrating p-values greater than 0.01 are considered statistically random [42]. The results of the NIST tests on the chaotic sequences generated by the 2D-CSHS are shown in Table 2. Both generated random sequences passed the tests, indicating that they possess good randomness.

## 3. Preliminary knowledge

### 3.1. ∞-shaped transformation

The space curve scan-fill transformation can effectively disrupt the order of elements in a two-dimensional matrix. To disturb the pixels of an image, an increasing number of space curve filling methods have been applied in image encryption [43], such as Zigzag transformation [44], Hilbert curves [45,46], sawtooth spiral lines [26], Y-index curves [47], L-shaped scanning [48], U-shaped scanning [24], and random order substitution [49]. However, these scanning methods primarily operate within a single pixel plane and fail to disrupt the inter-channel pixel relationships inherent in color images.

This paper proposes an ∞-shaped transformation filling method applied to a three-row matrix. A three-row matrix is chosen because color images are divided into three channels: R, G, and B. To disrupt the correlation between pixels across channels, each channel is converted into a one-dimensional vector and then merged by columns, resulting in a three-row matrix.

**Definition 1:** The ∞-shaped transformation is defined as follows:

$$\beta = P\alpha, \tag{6}$$

where $\alpha$ is the initial vector, $P$ is the permutation matrix which is invertible, and $\beta$ is the scrambled vector. Depending on the number of columns, permutation matrices of different sizes are selected.

To achieve dynamic scrambling, the ∞-shaped filling is divided into five cases based on the number of columns, each corresponding to a different permutation matrix. Taking the scrambling of a $3 \times 2$ matrix as an example, the detailed steps are described as follows:

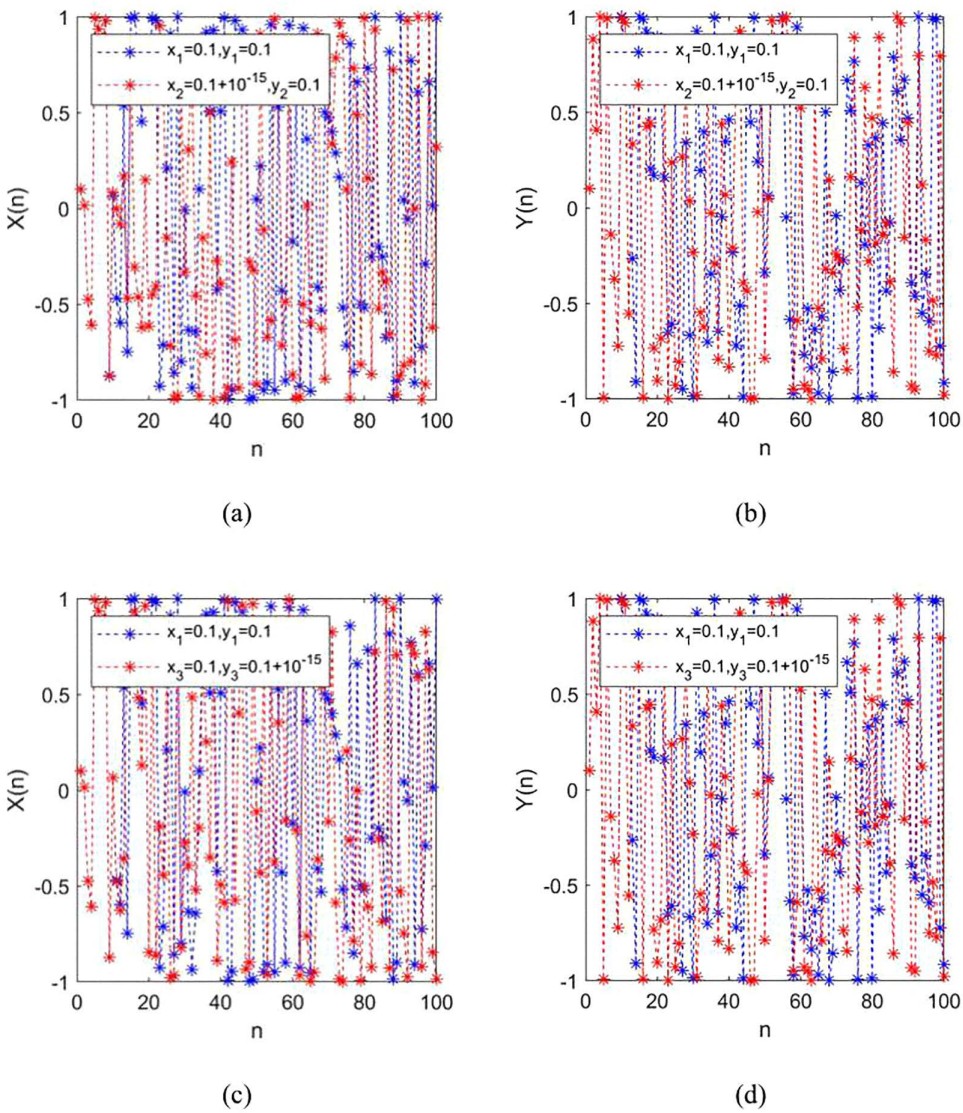

**Fig 7. Trajectory diagrams.** (a) Output $X$ of the 2D-CSHS under variation of the initial value $X$; (b) Output $Y$ of the 2D-CSHS under variation of the initial value $X$; (c) Output $X$ of the 2D-CSHS under variation of the initial value $Y$; (d) Output $Y$ of the 2D-CSHS under variation of the initial value $Y$.

[1] Convert the matrix $A$ into a column vector $\alpha$.

[2] Multiply $\alpha$ by a 6×6 permutation matrix $P$ on the left to obtain the scrambled matrix $\beta$, where

$$P = \begin{bmatrix} 0 & 0 & 0 & 0 & 1 & 0 \\ 0 & 0 & 0 & 1 & 0 & 0 \\ 1 & 0 & 0 & 0 & 0 & 0 \\ 0 & 1 & 0 & 0 & 0 & 0 \\ 0 & 0 & 0 & 0 & 0 & 1 \\ 0 & 0 & 1 & 0 & 0 & 0 \end{bmatrix}.$$

**Table 2. NIST test results for the proposed 2D-CSHS.**

| Subtests | X-seq | Result | Y-seq | Result |
|---|---|---|---|---|
| Frequency test | 0.4354 | Passed | 0.5313 | Passed |
| Block Frequency test | 0.7577 | Passed | 0.6815 | Passed |
| Runs Test | 0.2359 | Passed | 0.8584 | Passed |
| Longest-Run-of-Ones Test | 0.5773 | Passed | 0.3600 | Passed |
| Binary Matrix Rank Test | 0.0483 | Passed | 0.0325 | Passed |
| Discrete Fourier Transform Test | 0.5422 | Passed | 0.9768 | Passed |
| Non-Overlapping Matching Test | 0.5423 | Passed | 0.4432 | Passed |
| Overlapping Matching Test | 0.4137 | Passed | 0.6059 | Passed |
| Universal Statistical Test | 0.8766 | Passed | 0.5645 | Passed |
| Linear Complexity Test | 0.0848 | Passed | 0.8829 | Passed |
| Serial Test | 0.3805 | Passed | 0.8423 | Passed |
| Approximate Entropy Test | 0.8171 | Passed | 0.6368 | Passed |
| Cumulative Sums Test | 1.0000 | Passed | 0.8898 | Passed |
| Random Excursion Test | 0.3099 | Passed | 0.6617 | Passed |
| State −4 | 0.0720 | Passed | 0.6656 | Passed |
| State −3 | 0.2821 | Passed | 0.4081 | Passed |
| State −2 | 0.0865 | Passed | 0.8236 | Passed |
| State −1 | 0.1294 | Passed | 0.9909 | Passed |
| State 1 | 0.6259 | Passed | 0.7551 | Passed |
| State 2 | 0.5711 | Passed | 0.0643 | Passed |
| State 3 | 0.4036 | Passed | 0.7732 | Passed |
| Random Excursion Variant Test | 0.9249 | Passed | 0.7147 | Passed |
| State −4 | 0.6922 | Passed | 0.9601 | Passed |
| State −3 | 0.4468 | Passed | 0.8813 | Passed |
| State −2 | 0.3287 | Passed | 0.6264 | Passed |
| State −1 | 0.2494 | Passed | 0.5855 | Passed |
| State 1 | 0.1833 | Passed | 0.8135 | Passed |
| State 2 | 0.0910 | Passed | 0.9462 | Passed |
| State 3 | 0.0211 | Passed | 0.9901 | Passed |
| State 4 | 0.3693 | Passed | 0.8129 | Passed |

(3) Convert $\beta$ into a 3×2 matrix column-wise to obtain the scrambled matrix $B$.

Fig 8 illustrates the process of ∞-shaped transformation under different matrix dimensions. The traversal order of matrix elements varies with the number of columns, but the scanning path consistently aligns closely with the writing pattern of the ∞ symbol. Different column counts correspond to distinct matrix transformation results, effectively enhancing the complexity of the transformation.

To verify the effectiveness of the ∞-shaped transform compared to classical scrambling methods such as Zigzag, simulation experiments were conducted using a 256×256 "house" image, with the results presented in Table 3. Table 3 presents a comparative analysis of correlation coefficients for both adjacent pixels and inter-channel pixels under different scrambling schemes. It can be observed that the ∞-shaped transform outperforms the Hilbert transform, Zigzag transform, and Peano curve transform in disrupting both spatial correlations among neighboring pixels and statistical dependencies across color channels. Specifically, this scheme substantially reduces the horizontal, vertical, and diagonal correlations of

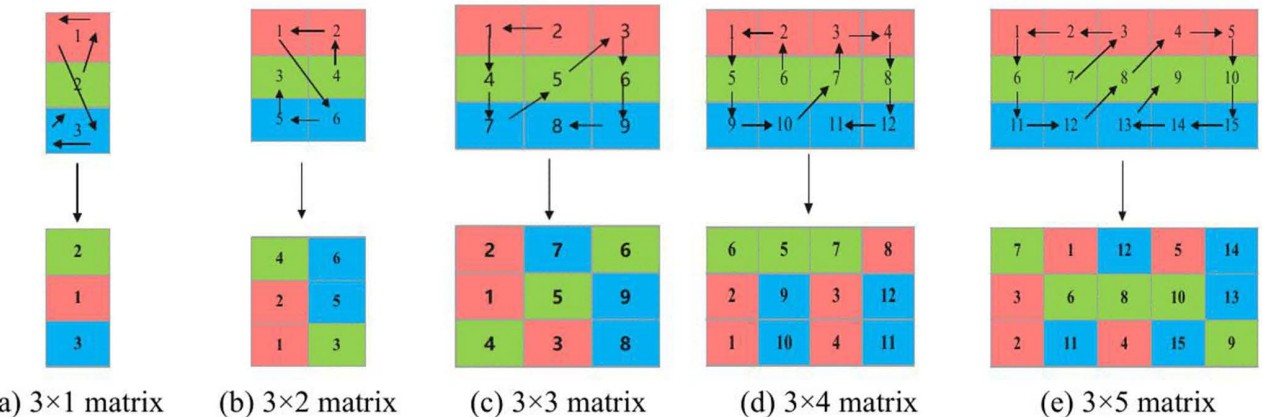

(a) 3×1 matrix  (b) 3×2 matrix  (c) 3×3 matrix  (d) 3×4 matrix  (e) 3×5 matrix

**Fig 8. Diagrams of the ∞-shaped transformation process.**

the R, G, and B channels, while also effectively weakening the inter-channel correlations (e.g., R–G, R–B, G–B). These results demonstrate the superior decorrelation capability and scrambling randomness of the ∞-shaped transform. In contrast, the other scrambling methods exhibit limited effectiveness in suppressing inter-channel correlation, with some cases showing negligible impact. Overall, the ∞-shaped transform achieves more comprehensive performance in image scrambling tasks.

### 3.2. Closed-loop control model

Traditionally, pixel scrambling and diffusion are executed as separate stages in image encryption. This paper establishes a closed-loop control model to increase the algorithm's complexity. The process begins by arranging a one-dimensional sequence into a ring structure, rotating clockwise. Subsequently, a random starting point is chosen for diffusion, which proceeds either clockwise or counterclockwise before the sequence is reverted to a one-dimensional format. This diffusion mechanism concurrently disrupts the sequence's order.

Let $A$ be a one-dimensional sequence of length m, with $n$ as the starting position ($n \leq m$). $X$ and *temp* represent random numbers, and $K$ denotes the key sequence. The encryption process involves the following steps:

**Step 1**: Constructing the closed loop

Form a closed loop $B$ by connecting the head and tail of vector $A$.

**Step 2**: Determining the encryption direction

Based on the random number $X$, decide whether to operate clockwise or counterclockwise on loop B.

**Step 3**: Executing the encryption process

Starting from the $n$-th element and using temp as the initial value: For clockwise operation, compute the transformed sequence $B'$ via Eq. (7). For counterclockwise operation, derive $B'$ using Eq. (8).

$$B'(i) = \begin{cases} \mathrm{mod}\left(B\left(\mathrm{mod}\left(n+i-1,m\right)+1\right) \oplus K(i) + temp, 256\right), i = 1 \\ \mathrm{mod}\left(B\left(\mathrm{mod}\left(n+i-1,m\right)+1\right) \oplus K(i) + B'(i-1), 256\right), i = 2 \\ \mathrm{mod}\left(B\left(\mathrm{mod}\left(n+i-1,m\right)+1\right) \oplus K(i) \oplus B'(i-1) + B'(i-2), 256\right), i > 2 \end{cases},$$

(7)

**Table 3. Comparison of correlation coefficients for different scrambling schemes.**

| Scrambling Path | | | Original image | Hilbert transform | Zigzag transform | Peano curve transform | proposed ∞-shaped transform |
|---|---|---|---|---|---|---|---|
| **Adjacent Pixel Correlation** | R | H | 0.9671 | 0.6984 | 0.1980 | 0.7679 | 0.6638 |
| | | V | 0.9353 | 0.6975 | 0.1977 | 0.6504 | 0.6696 |
| | | D | 0.9126 | 0.5613 | 0.1661 | 0.5504 | 0.5659 |
| | G | H | 0.9805 | 0.7614 | 0.1669 | 0.7899 | 0.7685 |
| | | V | 0.9474 | 0.7598 | 0.1674 | 0.7247 | 0.7643 |
| | | D | 0.9320 | 0.6367 | 0.1478 | 0.6240 | 0.5254 |
| | B | H | 0.9820 | 0.8190 | 0.2039 | 0.8404 | 0.7258 |
| | | V | 0.9749 | 0.8180 | 0.2055 | 0.7881 | 0.7318 |
| | | D | 0.9625 | 0.7101 | 0.2061 | 0.6903 | 0.7404 |
| **Inter-channel Pixel Correlation** | R-G | | 0.6378 | 0.6378 | 0.6378 | 0.6378 | 0.5420 |
| | R-B | | 0.4823 | 0.4823 | 0.4823 | 0.4823 | 0.6300 |
| | G-B | | 0.9418 | 0.9418 | 0.9418 | 0.9418 | 0.6499 |

$$B'(i) = \begin{cases} \mod\left(B\left(\mod\left(m+1-n-i, m\right)+1\right) \oplus K(i) + temp, 256\right), i = 1 \\ \mod\left(B\left(\mod\left(m+1-n-i, m\right)+1\right) \oplus K(i) + B'(i-1), 256\right), i = 2 \\ \mod\left(B\left(\mod\left(m+1-n-i, m\right)+1\right) \oplus K(i) \oplus B'(i-1) + B'(i-2), 256\right), i > 2 \end{cases}. \tag{8}$$

Fig 9 intuitively illustrates the operational process within the closed-loop control model. The closed-loop controlled diffusion process exhibits a strong directional dependence. When the identical key set {$K$, $n$, $temp$} is applied to the initial sequence $A$ = {35, 47, 59, 168, 205, 130, 20}, the output sequences are entirely distinct: {176, 188, 116, 31, 141, 2, 205} for clockwise diffusion versus {213, 61, 97, 118, 109, 100, 5} for its counterclockwise counterpart. This confirms that the operational direction is a decisive factor independent of the encryption keys.

**Step 4**: Obtaining the encrypted sequence

Unfold $B'$ to obtain the final encrypted sequence $C$.

## 4. Proposed encryption algorithm

This section presents a novel, efficient, and secure image encryption algorithm developed by integrating the 2D-CSHS for random matrix generation with an ∞-shaped transformation and closed-loop synchronous scrambling-diffusion. The comprehensive encryption framework is depicted in Fig 10. To facilitate clarity in the algorithmic description, we assume a target color image of dimensions $M \times N$.

### 4.1. Key generation

To mitigate plaintext attacks, SHA-384 is incorporated into the key generation process to produce plaintext-dependent algorithm keys. Initially, the target image information is processed through the SHA-384 function, generating a 96-digit hexadecimal hash value $h$= {$h(1)$, $h(2)$,…, $h(96)$}. Subsequently, this hash value sequence is divided into 24 groups according to Eq. (9):

$$k(i) = hex2dec(h((i\text{-}1) \times 4 + 1 : 4 \times i)). \tag{9}$$

To construct four sets of random sequences, two pairs of initial values {$x_{01}$, $y_{01}$} and {$x_{02}$, $y_{02}$} are input into the chaotic system for iteration. The detailed pseudocode for the key sequence generation process is provided in Algorithm 1.

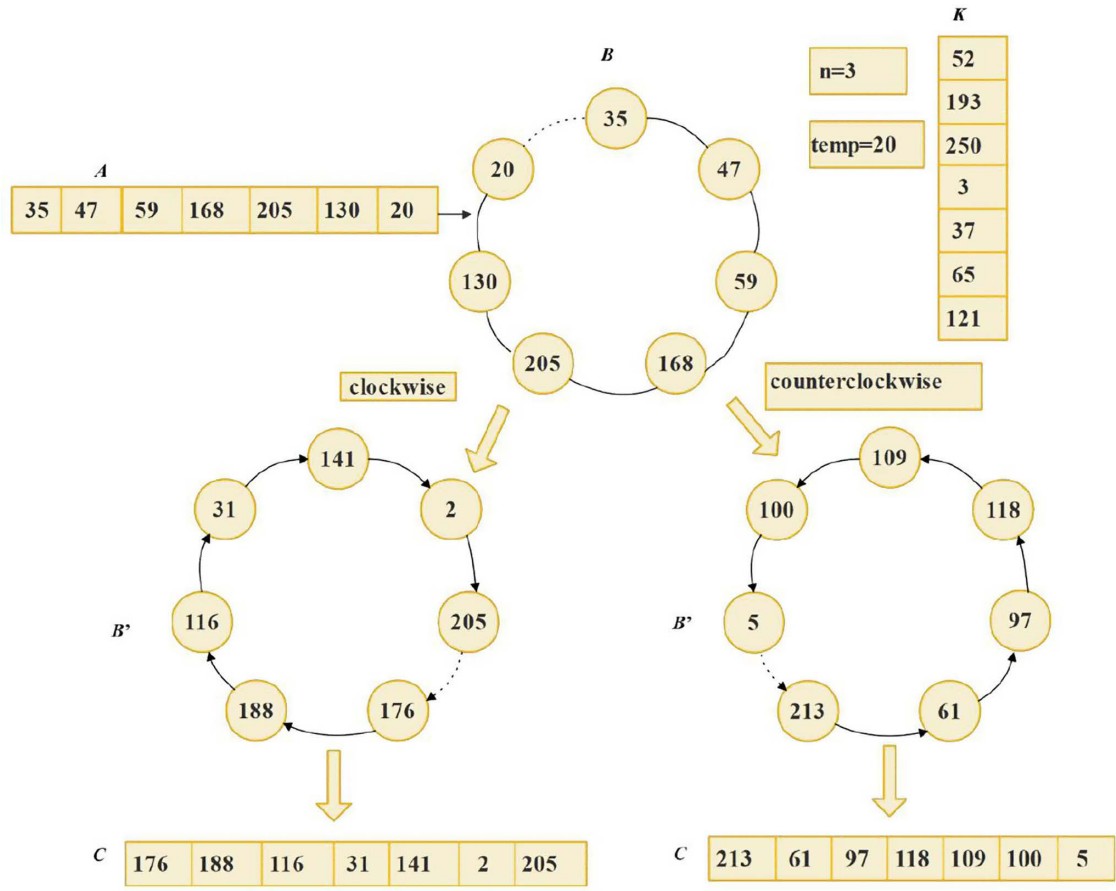

**Fig 9. Closed-loop synchronous scrambling-diffusion operation process.**

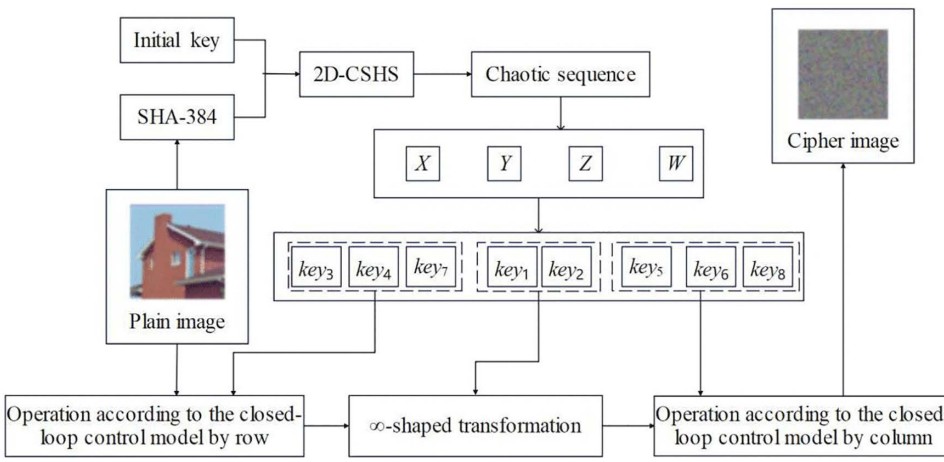

**Fig 10. Encryption process framework diagram.**

**Algorithm 1. Generation of chaotic random sequence**

```
Input: P, x₀₁, y₀₁, x₀₂, y₀₂
Output: key₁, key₂, key₃, key₄, key₅, key₆, key₇, key₈, temp₁, temp₂
1: [row col chan]← size(P)
2: N←row*col*chan
3: Nn←row*col
4: h←SHA384(P)
5: {k(1), k(2), ···,k(24)}←h
6:%Calculate the control parameters and initial values r₁, u₁,r₂, u₂ and x₁, y₁, x₂, y₂
7: r₁←5+mod((k(1)+ k(2)+ k(3))*2⁻¹⁰,5)
8: u₁← 1+mod((k(4)+ k(5)+ k(6))*2⁻¹⁰,9)
9: r₂← 5+mod((k(7)+ k(8)+ k(9))*2⁻¹⁰,5)
10: u₂← 1+mod((k(10)+ k(11)+ k(12))*2⁻¹⁰,9)
11: x₁← mod(x₀₁+(bitxor(k(13),k(14))+ k(15))/2¹⁶,1)
12: y₁← mod(y₀₁+(bitxor(k(16),k(17))+ k(18))/2¹⁶,1)
13: x₂← mod(x₀₂+(bitxor(k(19),k(20))+ k(21))/2¹⁶,1)
14: y₂← mod(y₀₂+(bitxor(k(22),k(23))+ k(24))/2¹⁶,1)
15:%Calculate the chaotic sequences X、Y、Z and W
16: [X,Y]←2D-CSHS(x₀₁, y₀₁, r₁, u₁, Nn+row+col*chan);
17: [Z,W]←2D-CSHS(x₀₂, y₀₂, r₂, u₂, N)
18:%Calculate the length of the sequence key₁
19: s=0; k_num=1;
20: while s <Nn && k_num <= length(X)
21: s←s+X(k_num)
22: k_num←k_num+1
23: end
24: k_num←k_num−1
25: if s>Nn
26: X (k_num)← Nn−(s− X(k_num))
27: end
28:%Calculate the key sequences key₁, key₂, key₃, key₄, key₅, key₆, key₇, key₈, temp₁, temp₂
28: key₁←X (1: k_num)
29: key₂←Y(1: k_num)
30: temp₁←mod(X(1+k_num)*10¹⁰, 256);
31: temp₂←mod(Y (1+k_num)*10¹⁰, 256);
32: key₃←X(Nn+1: Nn+row);
33: key₄←Y (Nn+1: Nn+row);
34: key₅←X (Nn+row+1: Nn+row+col* chan);
35: key₆←Y (Nn+row+1: Nn+row+col* chan);
36: key₇←mod(Z* 10¹⁵, 256);
37: key₈ ← mod(W* 10¹⁰, 256).
```

## 4.2. Encryption process

After generating the key sequence {$key_1$, $key_2$,..., $key_8$} from the input initial key and plain image information, the encryption operation can be further performed on the plain image $P$. The specific steps are described as follows:

**Step 1:** Concatenate the three channels of the color image matrix $P$ row-wise to form an $M \times 3N$ matrix $P_1$.

**Step 2:** For each row of matrix $P_1$, construct a closed-loop structure. The starting position in each loop is determined by $key_3$, while the diffusion direction is specified by the $key_4$ sequence. Using $temp_1$ and $key_7$ as diffusion keys, perform diffusion operations on each closed loop. The processed loops are then expanded and sequentially placed into the corresponding rows of matrix $P_2$.

**Step 3:** Reshape the image matrix $P_2$ into a $3 \times MN$ matrix $P_3$, and generate a corresponding matrix $P_4$ of the same dimensions.

**Step 4:** For matrix $P_3$, sequentially select column submatrices according to $key_1$, perform $\infty$-shaped transformation, then determine whether to pad them at the beginning or end of matrix $P_4$ based on $key_2$.

**Step 5:** The image matrix $P_4$ is then partitioned row-wise and reshaped into an $M \times 3N$ matrix $P_5$.

**Step 6:** For each column of matrix $P_5$, form a closed-loop structure. The starting position within each loop is determined by $key_5$, while the diffusion direction is governed by the $key_5$ sequence. Using $temp_2$ and $key_8$ as diffusion keys, perform diffusion operations on each closed-loop. The transformed loops are then expanded and sequentially arranged into the corresponding columns of matrix $P_6$.

**Step 7:** Reconstruct $P_6$ into an $M \times N$ color image, which constitutes the final encrypted image $C$.

Algorithm 2 outlines the pseudocode for the encryption procedure, incorporating two primary diffusion techniques: CL_diffusion, which denotes clockwise synchronous encryption diffusion, and RE_CL_diffusion, indicating counterclockwisesynchronous encryption diffusion.

### Algorithm 2. Encryption process

```
Input: P, X₁, Y₁, Z₁, Z₂, W₁, W₂, X₂, Y₂
Output: C
1:%Perform row-wise closed-loop synchronous scrambling-diffusion on image matrix P
2: Merge P row-wise into P₁
3: K₃←mod (round (key₃*10¹⁵), 3*col)+1
4: K₄←mod (floor (key₄*10¹⁵), 2)
5: temp←temp₁
6: K₇←reshape (key₇, row, 3*col)
7: for i←1 to row do
8:   A←P₁(i,:)
9:   K←K₇(i,:)
10:  if key₄(i)==0
11:    B←CL_diffusion (A, K₃ (i), temp, K, 3*col)
12:  else
13:    B←RE_CL_diffusion (A, K₃(i), temp, K, 3*col)
14:  end
15:  temp←B(3*col)
16:  P₂(i,:) ← B
17: end
18:%∞-shaped transformation for image matrix scrambling
19: Reshape P₂ column-wise into a 3-row by row*col-column matrix P₃
20: n←0; k₁←0; k₂←0;
21: for j←1 to k_num do
22:   A←P₃ (:, n+1: n+key₁(j))
23:   B←transformation (A, key₁ (j))
24:   if key₂(j) == 0
25:     P₄(:,1+k1:k1+key1(j)) ← B
26:     k₁ ← k₁+ key₁(j)
27:   else
28:     P₄(:, Nn- k₂- key₁(j)+1: Nn- k₂)← B
29:     k₂ ← k₂ + key₁(j)
30:   end
31:   n←n+key₁(j)
32: end
33:%Perform column-wise closed-loop synchronous scrambling-diffusion on image matrix P₄
34: Merge image matrix P₄ row-wise to reconstruct image P₅
35: temp←temp₂
36: K₈←reshape (key₈, row, 3*col)
37: for j←1:3*col do
```

```
38: A←P₅(:, j)
39: K←K₈(:, j)
40: if key₆(j) ==0
41: B←CL_diffusion(A, key₅(j), temp, K, row)
42: else
43: B←RE_CL_diffusion(A, key₅(j), temp, K, row)
44: end
45: temp← B(row)
46: P₆ (:, j)← B
47: end
48: Reconstruct matrix P₆ into the color encrypted image C
```

### 4.3. Decryption algorithm

The proposed algorithm in this paper is a symmetric encryption algorithm, whose decryption is the inverse process of encryption. During decryption, the key, and the ciphertext image are all transmitted to the recipient. The specific steps are described as follows:

**Step 1:** Convert the ciphertext image $C$ into an $M \times 3N$ image matrix $I_6$.

**Step 2:** Connect each column of matrix $I_6$ into a closed loop, and perform inverse closed-loop diffusion on each using the sequences $key_5$, $key_6$, $key_8$, and $temp_2$ to obtain new closed-loop data.

**Step 3:** Construct an empty $M \times 3N$ matrix $I_5$, then expand the closed loops obtained in Step 2 and fill them column-wise into matrix $I_5$.

**Step 4:** Convert $I_5$ into a $3 \times MN$ matrix $I_4$.

**Step 5:** Perform an inverse $\infty$-shaped transformation on $I_4$ using $key_1$ and $key_2$ to obtain matrix $I_3$.

S**tep 6:** Reshape $I_3$ into an $M \times 3N$ matrix $I_2$. Then, connect each row of $I_2$ into a closed loop and apply inverse closed-loop diffusion using $key_3$, $key_4$, $key_7$, and $temp_1$ to derive new closed-loop data.

**Step 7:** Expand each new closed loop obtained above and fill them row-wise back into $I_2$, resulting in a new matrix $I_1$.

**Step 8:** Convert $I_1$ into an $M \times N \times 3$ color image matrix $I$, which is the decrypted image.

## 5. Experiments and analysis

We conducted simulation experiments and algorithm analysis on the proposed scheme using MATLAB R2022a software. The experimental environment was Windows 10, with a computer configuration of an Intel(R) Core(TM) i5-8265U CPU @ 1.60GHz (1.80 GHz) and 8GB RAM. To verify the algorithm effectiveness, four distinct color images were used for simulation experiments: "House" (256×256), "Baboon" (512×512), "Peppers" (512×512), and "SanDiego" (1024×1024). The encryption performance was analyzed, and the encryption and decryption results are shown in Fig 11.

### 5.1. Key space analysis

The secret key of the proposed algorithm consists of two parts: a 384-bit hash value $h$ generated from the plaintext image, and a set of initial values and system parameters for the chaotic maps. The overall key space must be evaluated based on the effective entropy provided by each key component, considering the practical limitations of computer representation. According to the IEEE 754 double-precision floating-point standard, the mantissa provides approximately 52 bits of effective precision per value. Therefore, each floating-point number (initial value or parameter) contributes at most about $2^{52}$ distinct possibilities in a computational environment.

We calculate the key space as follows: (1) It comprises the hash $h$ (384 bits) and four initial values. Each initial value provides about 52 bits of entropy. Thus, the total effective key space is approximately: $2^{384} \times (2^{52})^4 = 2^{592}$. (2) It comprises eight floating-point numbers (four initial values and four parameters). Each contributes about 52 bits of entropy, yielding: $(2^{52})^8 = 2^{416}$. Since an attacker may target the weaker of the two key forms, the effective key space of our algorithm

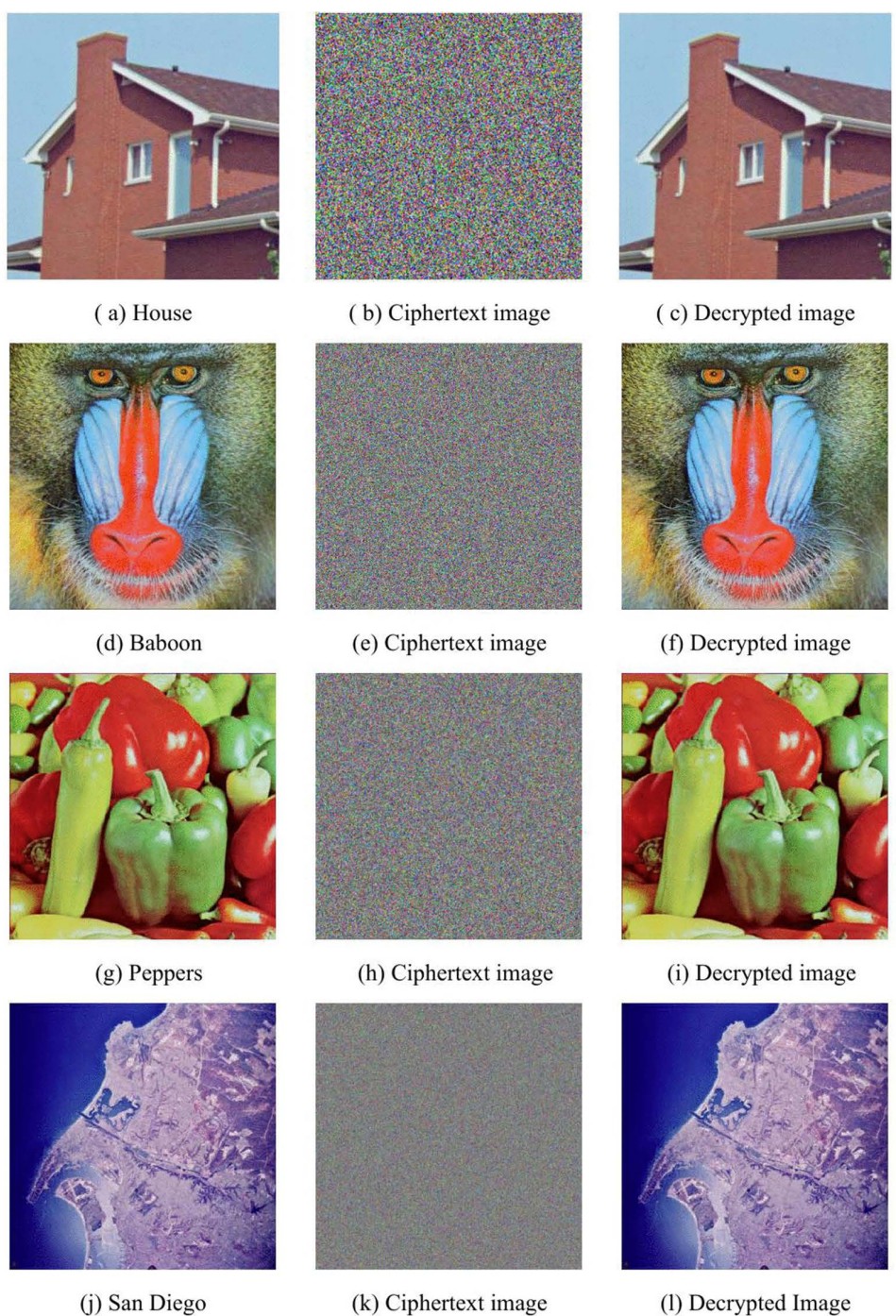

**Fig 11. Simulation results of encryption and decryption for different color images.**

is determined by the smaller of the two estimates, which is $2^{416}$. This key space size of $2^{416} \approx 1.04 \times 10^{125}$ far exceeds the minimum requirement of $2^{100}$ to resist brute-force attacks Even considering potential security margins and the theoretical limitations of chaotic map implementations, the key space remains sufficiently large to ensure practical security. A comparison of the key spaces between the proposed algorithm and those in references [7,50], and [51] is present in Table 4.

**Table 4.  Comparison of key space.**

| Algorithm | Ref. [7] | Ref. [50] | Ref. [51] | Proposed Algorithm |
|---|---|---|---|---|
| Key space | $2^{256}$ | $2^{256}$ | $2^{142}$ | $2^{416}$ |

## 5.2.  Key sensitivity analysis

A highly secure encryption system must exhibit extreme sensitivity to its secret keys; even the slightest modification should render the decrypted image incorrect or completely unrecognizable. To verify the key sensitivity of our encryption system, Fig 12 presents the decryption results using the original keys ($x_{01}$, $x_{02}$, $y_{01}$, $y_{02}$), and modified keys where each original key was perturbed by $10^{-15}$. The experimental results demonstrate that any minor alteration to the original keys produces a completely unrecognizable decrypted image, devoid of meaningful information from the plaintext image. This indicates that the proposed algorithm is highly sensitive to key variations, thereby significantly enhancing its overall security.

## 5.3.  Chosen-plaintext attack analysis

A robust encryption algorithm must resist chosen-plaintext attacks, meaning that even fully black or fully white images should produce ciphertext results that are entirely unrecognizable. To validate this resistance, we encrypted both black

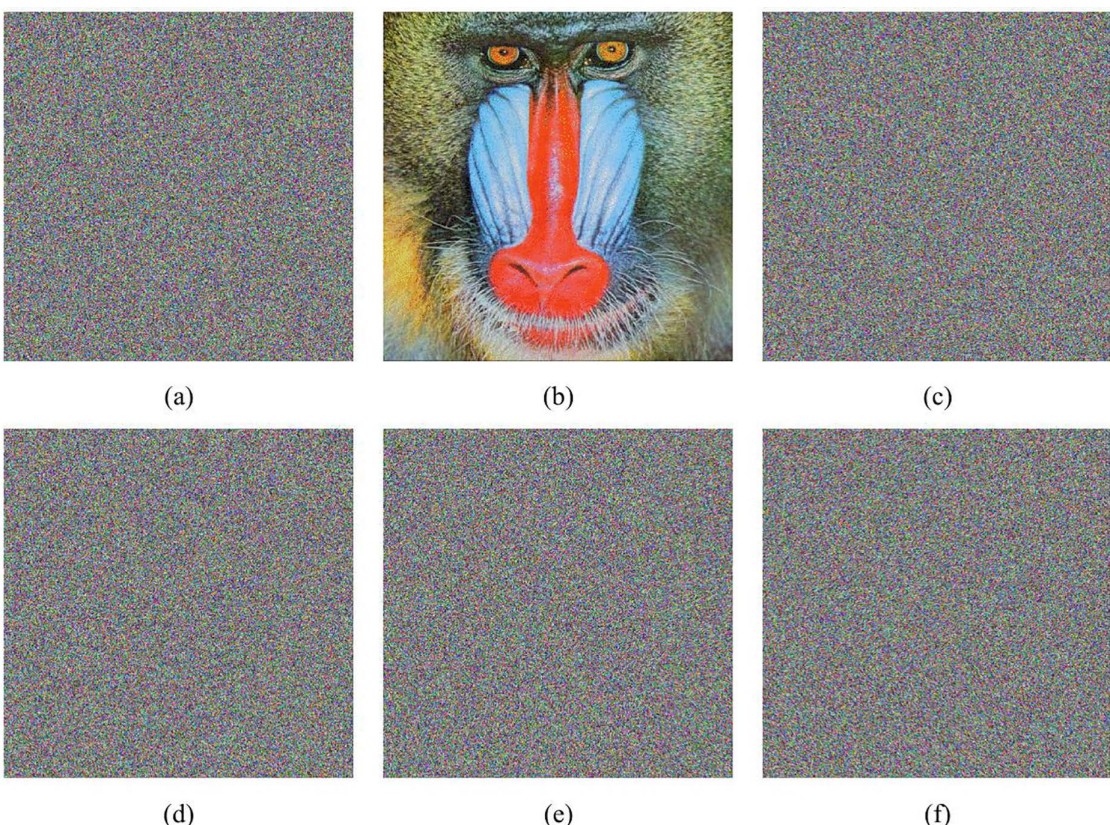

(a)   (b)   (c)

(d)   (e)   (f)

**Fig 12.  Key sensitivity analysis.** (a) Ciphertext image encrypted with the original key; (b) Decrypted image with the original key; (c) Decryped image using $x_1 + 10^{-15}$; (d) Decrypted image using $y_1 + 10^{-15}$; (e) Decrypted image using $x_2 + 10^{-15}$; (f) Decrypted image using $y_2 + 10^{-15}$.

and white images separately. The results are shown in Fig 13. As observed, the ciphertext images retain no discernible information from the original plaintext images. Thus, the proposed algorithm demonstrates strong resistance against chosen-plaintext attacks.

## 5.4. Histogram analysis

Images with meaning often display unique pixel distribution traits. To mask these statistical signatures in plaintext images and thwart statistical attack methodologies, a robust encryption scheme is required to alter the ciphered image's pixel distribution towards uniformity. Fig 14 contrasts the histograms of four sample images in their plaintext and ciphertext states.

Upon visual examination, it is evident that the ciphertext images exhibit a near-uniform spread of pixel intensity values. This observation substantiates the efficacy of the encryption procedure in accomplishing the following objectives: normalizing the distribution of pixel values,erasing the inherent statistical structure of the original data,and fortifying the image against statistical cryptanalysis attempts.

To objectively evaluate the uniformity of pixel distribution in the encrypted image, we also performed a chi-square test. For 8-bit images, the chi-square value is calculated as shown in Eq. (10).

$$\chi^2 = \sum_{i=1}^{256} \frac{(P_i - M \times N \times 1/256)^2}{M \times N \times 1/256}$$

(10)

where $P_i$ is the number of pixels with the value of $i-1$, and $M$ and $N$ correspond to the number of rows and the number of columns in an image, respectively. At a significance level of 0.05, the chi-square distribution yields a critical value of 293.2478 [6]. We conducted chi-square tests on each channel of the four encrypted images, with the results presented in Table 5. All images passed the chi-square test, indicating that the encrypted images generated by the proposed algorithm exhibit a high degree of pixel distribution uniformity. This demonstrates the algorithm's robustness against statistical analysis attacks targeting pixel values.

## 5.5. Correlation analysis

Natural images typically exhibit strong correlations between adjacent pixels due to their inherent spatial continuity. To prevent potential attackers from deducing original image information via correlation analysis, an effective encryption algorithm must significantly reduce such inter-pixel dependencies.

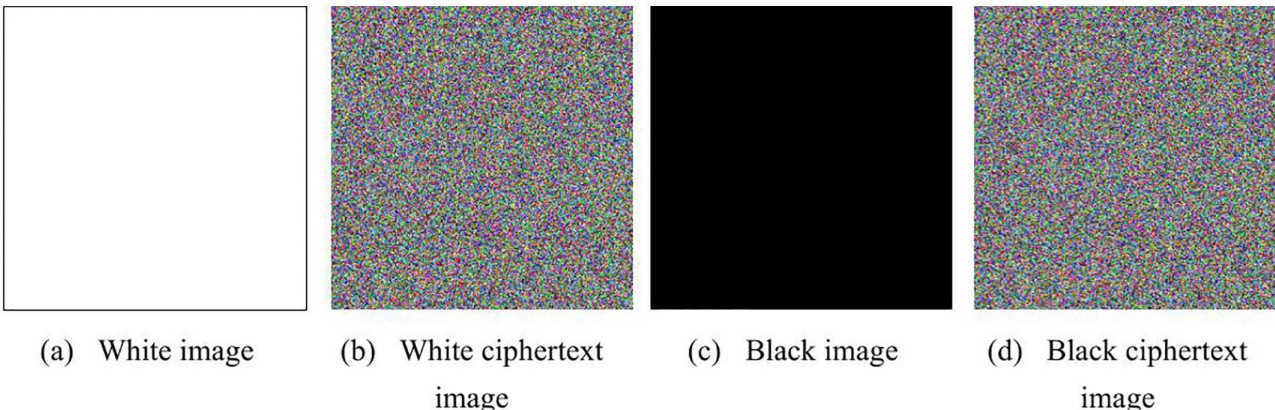

(a)  White image  (b)  White ciphertext image  (c)  Black image  (d)  Black ciphertext image

**Fig 13. Chosen-plaintext attack results.** (a) Plaintext White image; (b) Ciphertext of White image; (c) Plaintext Black image; (d) Ciphertext of Black image.

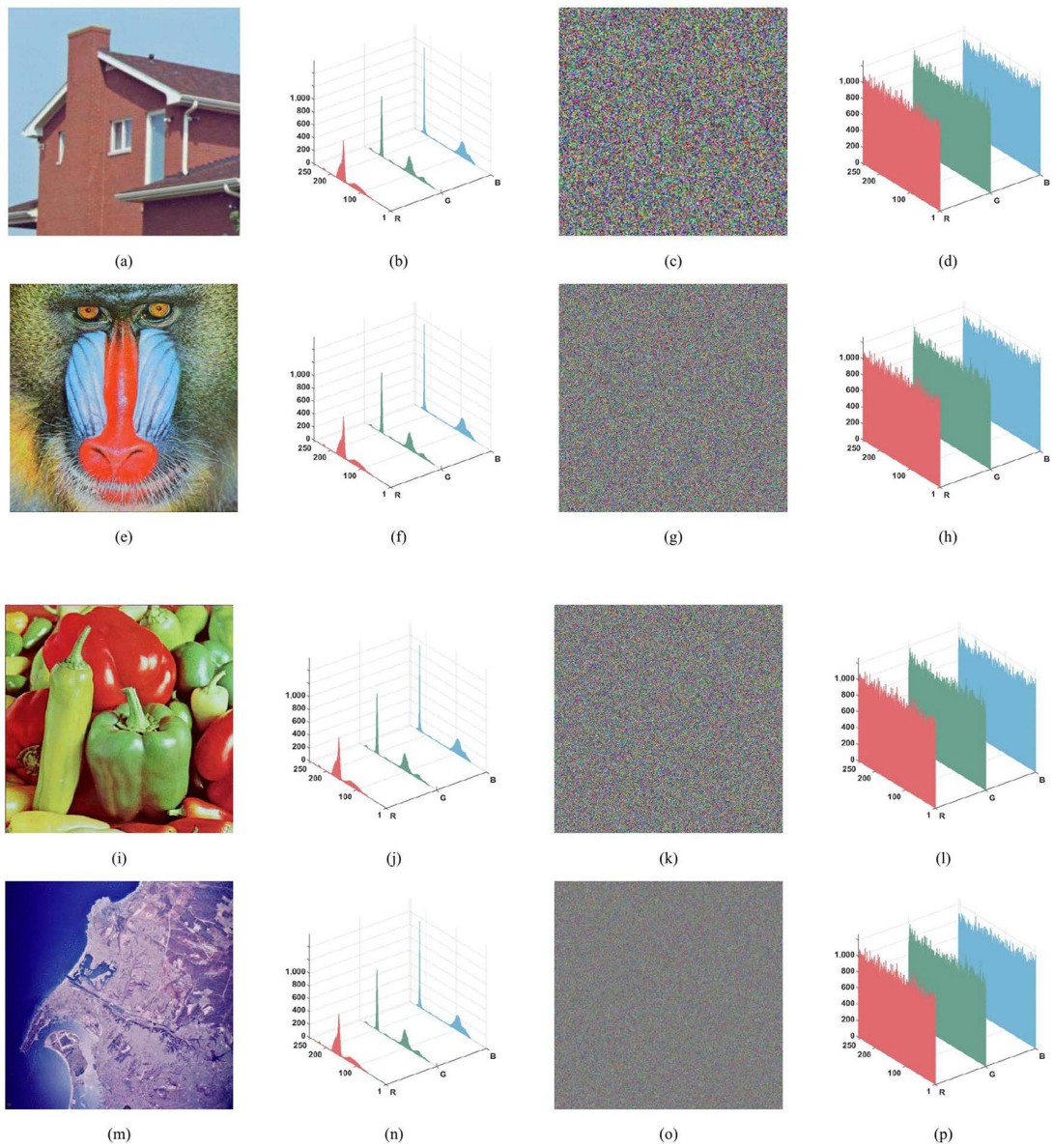

**Fig 14. Plaintext and ciphertext histograms of different color images.** (a) House; (b) Histogram of House; (c) Ciphertext House;(d) Histogram of ciphertext House; (e) Baboon; (f) Histogram of Baboon; (g) Ciphertext Baboon; (h) histogram of Ciphertext Baboon; (i) Peppers; (j) Histogram of Peppers; (k) Ciphertext Peppers; (l) Histogram of ciphertext Peppers; (m) San Diego; (n) Histogram of San Diego; (o) Ciphertext San Diego; (p) Histogram of ciphertext San Diego.

Meaningful images typically exhibit strong correlations between adjacent pixels, whereas cihpertext images demonstrate significantly reduced pixel correlations in the spatial domain. The correlation coefficient is an effective metric for quantifying the linear dependence between neighboring pixels along specified directions. Key characteristics of correlation coefficients include values approaching ±1 (indicating strong correlations) and a coefficient of 0 (indicating no linear relationship). The analysis focuses on three principal directions: horizontal, vertical, and diagonal.

**Table 5. Chi-square test results.**

| Size | Image | Channel | χ² Value (<293.2478) | Result |
|------|-------|---------|---------------------|--------|
| 256×256 | House | R | 254.6406 | Passed |
| | | G | 253.9922 | Passed |
| | | B | 266.4609 | Passed |
| 512×512 | Peppers | R | 252.2090 | Passed |
| | | G | 261.6777 | Passed |
| | | B | 228.9180 | Passed |
| 512×512 | Baboon | R | 259.9727 | Passed |
| | | G | 229.3867 | Passed |
| | | B | 291.7598 | Passed |
| 1024×1024 | San Diego | R | 253.3315 | Passed |
| | | G | 221.8442 | Passed |
| | | B | 214.1904 | Passed |

For a given image, $n$ pairs of adjacent pixels are randomly selected and denoted as $(x, y)$. The correlation coefficients are calculated by Eqs. (11)-(14) as follows:

$$\rho_{xy} = \frac{\text{cov}(x, y)}{\sqrt{D(x)}\sqrt{D(y)}},$$
(11)

$$\text{cov}(x, y) = \frac{1}{n}\sum_{i=1}^{n}(x_i - E(x))(y_i - E(y)),$$
(12)

$$D(x) = \frac{1}{n}\sum_{i=1}^{n}(x_i - E(x))^2,$$
(13)

$$E(x) = \frac{1}{n}\sum_{i=1}^{n}x_i .$$
(14)

Table 6 presents the correlation coefficients of adjacent pixels along three directions for four test images, both before and after encryption. The original images exhibit strong correlations in all directions, while the ciphertext images show correlation coefficients near zero, demonstrating excellent encryption performance.

This quantitative analysis confirms the algorithm's effectiveness in breaking spatial correlations, achieving near-ideal decorrelation performance that meets strict cryptographic requirements. The scatter plots in Fig 15 provide visual evidence of this decorrelation effect, showing randomized pixel distributions in all three orientations after encryption.

### 5.6. Information entropy

Information entropy serves as a crucial metric for quantifying randomness. In image encryption, it is employed to assess the degree of randomness in digital image data, reflecting the uncertainty of image pixels [36]. For an information source $m$, the information entropy is calculated using Eq. (15).

**Table 6. Correlation coefficients of different images.**

| Algorithms | Images | | Horizontal | Vertical | Diagona |
|---|---|---|---|---|---|
| **Proposed** | **House** | R | −0.0078 | 0.00005 | −0.0042 |
| | | G | 0.0007 | −0.0040 | −0.0049 |
| | | B | −0.0045 | −0.0045 | −0.0005 |
| | **Peppers** | R | −0.0063 | −0.0081 | 0.0006 |
| | | G | 0.00009 | 0.0030 | −0.0011 |
| | | B | 0.0047 | −0.0065 | −0.0006 |
| | **Baboon** | R | −0.0007 | 0.0006 | 0.0057 |
| | | G | 0.0001 | −0.0005 | 0.0059 |
| | | B | 0.0047 | 0.0004 | 0.0070 |
| | **San Diego** | R | 0.0040 | 0.0005 | 0.0006 |
| | | G | 0.0020 | −0.0004 | −0.0029 |
| | | B | −0.0068 | −0.0091 | −0.0008 |
| Ref. [3] | Peppers | R | 0.0009 | −0.0044 | −0.0048 |
| | | G | 0.0097 | 0.0014 | −0.0001 |
| | | B | 0.0021 | 0.0064 | 0.0002 |
| | Baboon | R | −0.0039 | 0.0018 | 0.0012 |
| | | G | −0.0020 | 0.0007 | −0.0019 |
| | | B | −0.0078 | −0.0032 | 0.0073 |
| Ref. [34] | Peppers | R | −0.0008 | −0.0014 | −0.0009 |
| | | G | 0.0016 | −0.0012 | −0.0000 |
| | | B | 0.0004 | −0.0009 | 0.0008 |
| | San Diego | R | 0.0003 | −0.0018 | −0.0003 |
| | | G | −0.0007 | 0.0004 | 0.0005 |
| | | B | 0.0011 | 0.0005 | 0.0011 |
| Ref. [52] | Peppers | R | 0.0008 | 0.0013 | 0.0037 |
| | | G | 0.0028 | 0.0021 | 0.0007 |
| | | B | 0.0014 | 0.0017 | 0.0003 |
| | Baboon | R | −0.0002 | 0.0010 | −0.0001 |
| | | G | 0.0008 | −0.0017 | −0,0016 |
| | | B | 0.0017 | 0.0010 | 0.0016 |
| Ref. [53] | Peppers | R | 0.0014 | 0.0025 | −0.00009 |
| | | G | 0.0008 | 0.0019 | −0.0034 |
| | | B | 0.0012 | −0.0003 | 0.0041 |
| Ref. [54] | Peppers | R | −0.0027 | −0.0048 | 0.0053 |
| | | G | 0.0004 | 0.0029 | −0.0020 |
| | | B | −0.0047 | 0.0019 | 0.0079 |

$$H(m) = -\sum_{i=1}^{L} p(m_i) \log_2 p(m_i),$$

(15)

where $p(m_i)$ is the frequency of signal $m_i$, and $L$ is the total number of distinct $m_i$. A higher information entropy indicates better randomness in the image. For an image with 256 grayscale levels, the ideal information entropy for an encrypted image is 8. Table 7 presents the RGB entropy values before and after image encryption. The results show that the entropy values of the cihpertext images are all close to 8, indicating strong randomness in the encrypted data.

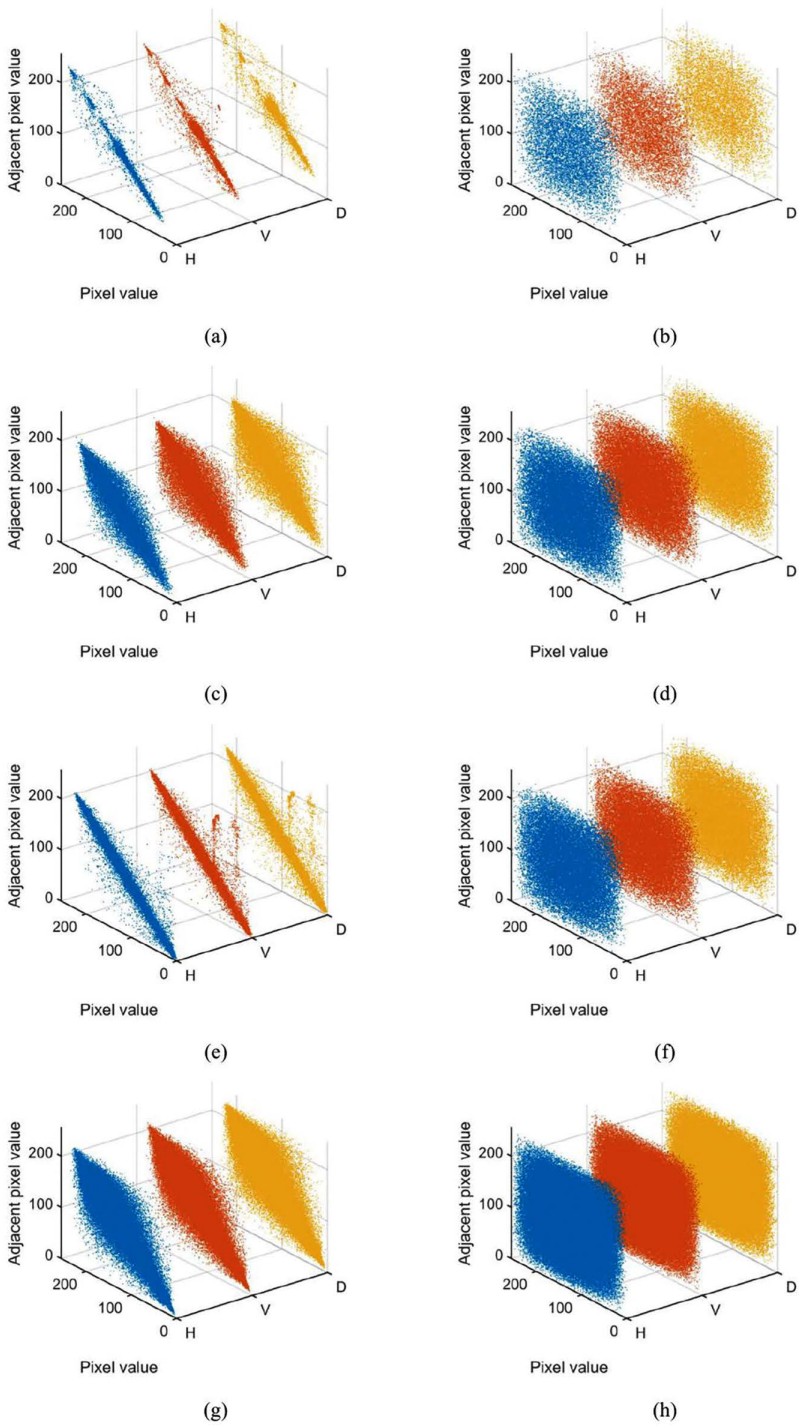

**Fig 15. Pixel correlation between the plaintext and ciphertext images invarious direction (left column: plaintext images; right column: ciphertext images; top to bottom: House, Baboon, Peppers, San Diego).**

**Table 7. Comparison of information entropy between plaintext and ciphertext images.**

| Algorithms | Images | Plain image | | | Encrypted image | | |
|---|---|---|---|---|---|---|---|
| | | R | G | B | R | G | B |
| Proposed | House | 6.4311 | 6.5389 | 6.2320 | 7.9969 | 7.9970 | 7.9975 |
| | Baboon | 7.7067 | 7.4744 | 7.7522 | 7.9993 | 7.9994 | 7.9994 |
| | Peppers | 7.3388 | 7.3388 | 7.0583 | 7.9993 | 7.9993 | 7.9993 |
| | San Diego | 7.7575 | 7.3387 | 6.9561 | 7.9998 | 7.9998 | 7.9999 |
| Ref. [3] | Baboon | 7.7067 | 7.4744 | 7.7522 | 7.9992 | 7.9993 | 7.9994 |
| | Peppers | 7.3388 | 7.3388 | 7.0583 | 7.9994 | 7.9993 | 7.9993 |
| Ref. [28] | Peppers | 7.3388 | 7.3388 | 7.0583 | 7.9993 | 7.9992 | 7.9993 |
| Ref. [51] | Peppers | 7.3388 | 7.4963 | 7.0583 | 7.9992 | 7.9993 | 7.9993 |
| Ref. [53] | Baboon | 7.6191 | 7.3817 | 7.6864 | 7.9993 | 7.9993 | 7.9993 |
| | Peppers | 7.3084 | 7.5584 | 7.0960 | 7.9992 | 7.9992 | 7.9993 |
| Ref. [55] | Baboon | 7.6191 | 7.3817 | 7.6864 | 7.9974 | 7.9969 | 7.9971 |
| | Peppers | 7.3084 | 7.5584 | 7.0960 | 7.9968 | 7.9972 | 7.9974 |

## 5.7. Differential attack analysis

A differential attack constitutes a form of chosen-plaintext attack. In this approach, the attacker encrypts pairs of plain images differing marginally and subsequently examines variations in their resultant ciphertexts to infer critical details about the encryption mechanism. Consequently, the effectiveness of such an attack hinges on whether minute alterations in the original image induce substantial discrepancies in the encrypted output. To evaluate plaintext sensitivity, the Number of Pixels Change Rate (NPCR) and the Unified Average Changing Intensity (UACI) are employed as two pivotal metrics [53]. These metrics are formally defined by Eqs. (16)-(17).

$$NPCR = \frac{1}{M \times N} \sum_{i=1}^{M} \sum_{i=1}^{N} D(i,j) \times 100\%,$$

$$(16)$$

$$UACI = \frac{1}{M \times N} \sum_{i=1}^{M} \sum_{i=1}^{N} \frac{|C_1(i,j) - C_2(i,j)|}{255} \times 100\%,$$

$$(17)$$

where $D(i,j) = \begin{cases} 0 & C_1(i,j) = C_2(i,j) \\ 1 & else \end{cases}$.

To validate the statistical significance of the results, hypothesis tests were conducted at a predetermined significance level. It should be noted that the critical threshold for *NPCR* and the acceptance interval for *UACI* vary depending on the image size and the chosen significance level. Taking $\alpha = 0.05$ as an example, the *NPCR* critical threshold ($N_{0.05}$) and the *UACI* acceptance intervals ([$U_{0.05}^{-}$, $U_{0.05}^{+}$]) for commonly used image sizes in experiments are shown in Table 8. Importantly, in randomness testing, the acceptance interval for $\alpha = 0.05$ is narrower (i.e., stricter) than that for $\alpha = 0.01$, requiring experimental values to be closer to the theoretical ideal [56].

After slightly changing the pixel values of the plaintext image, we separately encrypted both the changed image and the plaintext image, and calculated their *NPCR* and *UACI* values. The results are presented in Table 9. As shown, all *NPCR* values exceed the respective thresholds, while all *UACI* values fall strictly within the theoretically acceptable ranges. This confirms that the encryption algorithm passes the statistical test at a 5% significance level.

**Table 8. NPCR critical threshold and UACI acceptance interval (%, α = 0.05) [56].**

| Size | $N^*_{0.05}$ | UACI acceptance interval |
|---|---|---|
| 256 × 256 | 99.5693 | [33.2824, 33.6447] |
| 512 × 512 | 99.5893 | [33.3730, 33.5541] |
| 1024 × 1024 | 99.5994 | [33.4183, 33.5088] |

**Table 9. NPCR and UACI values (%).**

| Algorithm | Image | Size | NPCR | UACI |
|---|---|---|---|---|
| **Proposed** | House | 256 × 256 | 99.6099 | 33.4538 |
| | Baboon | 512 × 512 | 99.5884 | 33.4710 |
| | Peppers | 512 × 512 | 99.6104 | 33.4682 |
| | San Diego | 1024 × 1024 | 99.6103 | 33.4678 |

## 5.8. Cropping and noise attacks

A robust encryption algorithm must exhibit resilience, capable of restoring image data despite malicious modifications to ciphertext during transmission, such as cropping or noise injection. Fig 16 depicts the decryption outcomes for baboon ciphertexts under diverse intensities of salt-and-pepper noise, consistently revealing the underlying image content.

Figs 17(a)-17(d) display ciphertexts undergoing various cropping assaults, alongside their decrypted counterparts in Figs 17(e)-17(h). Remarkably, even with a 40% reduction in ciphertext pixels due to cropping, the decryption mechanism effectively retrieves critical image features. This evidence confirms the algorithm's capability to withstand noise disruptions and cropping intrusions, demonstrating its high robustness.

## 5.9. Peak signal-to-noise ratio analysis

Although the decrypted images under cropping and noise attacks are visually recognizable, the subjective evaluation is not accurate enough. The peak signal-to-noise ratio (PSNR) can provide a more objective assessment of image quality. PSNR quantifies the image distortion by calculating the mean square error (MSE) between the original image and the image to be evaluated. It is defined as Eq. (18).

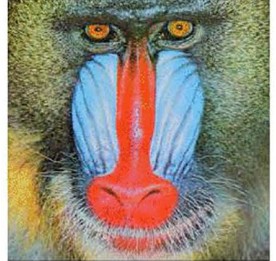 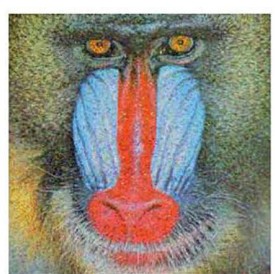 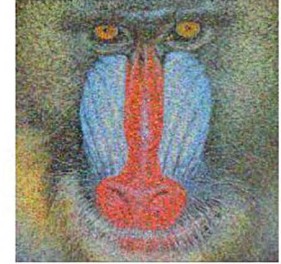 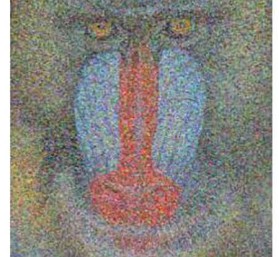

(a) Decrypted Baboon without noise (b) Decrypted Baboon with 0.02 noise density (c) Decrypted Baboon with 0.05 noise density (d) Decrypted Baboon with 0.1 noise density

**Fig 16. Results of anti-noise attack simulations. (a)** Decrypted Baboon without noise; **(b)** Decrypted Baboon with 0.02 noise density; **(c)** Decrypted Baboon with 0.05 noise density; **(d)** Decrypted Baboon with 0.1 noise density.

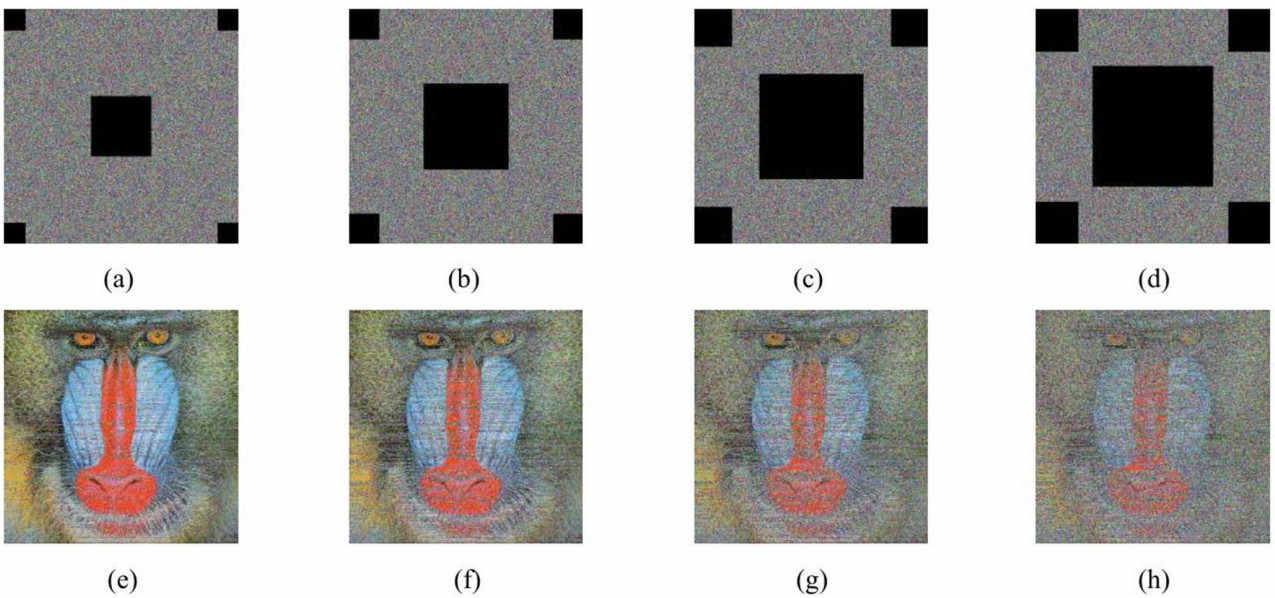

**Fig 17. Cropping attack analyses of Baboon.** (a) Ciphertext with crop 10%; (b) Ciphertext with crop 20%; (c) Ciphertext with crop 30%; (d) Ciphertext with crop 40%; (e) Decrypted image of (a); (f) Decrypted image of (b); (g) Decrypted image of (c); (h) Decrypted image of (d).

$$\begin{cases} PSNR = 10\lg\left(\frac{MAX_I^2}{MSE}\right) \\ MSE = \frac{1}{M \times N}\sum_{i=1}^{M}\sum_{j=1}^{N}\left(I\left(i,j\right) - C\left(i,j\right)\right)^2 \end{cases},$$

(18)

where MSE denotes the mean squared error between the plain image $I$ and the cipher image $C$. The height and width of the image are represented by $M$ and $N$, respectively. $MAX_I$ is the maximum possible pixel value of the image (for an 8-bit grayscale image, the maximum value is 255).

The PSNR values between the decrypted images (after cropping and noise addition) and the original images are shown in Tables 10 and 11. As observed, the PSNR values remain within acceptable ranges even under severe cropping ratios and high noise intensities. This confirms the excellent robustness of the proposed algorithm.

## 5.10. Algorithm complexity analysis

The time complexity of the proposed encryption algorithm is primarily determined by its iterative and transformation operations. For a color plaintext image of size $M \times N$, the core components, including chaotic sequence generation (where $L \propto M \times N$), row/column closed-loop diffusion, $\infty\infty$-shaped transformation, and matrix reorganization, each process every pixel exactly once. Consequently, all these stages achieve a time complexity of $O(M \times N)$.

In summary, the overall time complexity of the encryption scheme is $O(M \times N)$, which corresponds to linear time complexity with respect to the total number of pixels. This indicates that the proposed algorithm has good scalability and can efficiently handle high-resolution images.

**Table 10. PSNR vaules under cropping attacks.**

| Image | PSNR (dB) | | | |
|---|---|---|---|---|
| | crop 10% | crop 20% | crop 30% | crop 40% |
| House | 15.0142 | 12.4646 | 11.0952 | 10.1179 |
| Baboon | 14.9125 | 12.3875 | 10.9672 | 10.0405 |
| Peppers | 14.1719 | 11.6481 | 10.2363 | 9.3386 |
| San Diego | 18.4360 | 15.2502 | 13.6020 | 12.4372 |

**Table 11. PSNR values under salt-and-pepper noise attacks.**

| Image | PSNR (dB) | | |
|---|---|---|---|
| | 0.02 | 0.05 | 0.1 |
| House | 16.7153 | 13.1998 | 11.0554 |
| Baboon | 16.5854 | 13.0878 | 13.1244 |
| Peppers | 15.8672 | 12.4386 | 10.2039 |
| San Diego | 16.2560 | 12.7913 | 10.6101 |

## 5.11. Time efficiency analysis

In addition to security analysis, time efficiency is a crucial metric for evaluating encryption algorithm performance. Table 12 displays the encryption and decryption times for images of varying sizes using the proposed algorithm. Note that actual computational efficiency may vary depending on hardware specifications and implementation details.

## 6. Conclusion and outlook

This paper introduces a novel two-dimensional hyperchaotic map (2D-CSHS), constructed through a nonlinear combination of sine and cosine terms derived from an improved cubic map. Its chaotic characteristics are rigorously validated using phase diagrams, LEs, SE, and sensitivity analysis, with comparative evaluations demonstrating significant advantages, including a wide chaotic range, high LE values, and uniform trajectory distribution. Leveraging this chaotic source, we propose an innovative color image encryption scheme that integrates an $\infty$-shaped pixel perturbation method with a closed-loop synchronous scrambling-diffusion mechanism. The scheme generates initial keys from the original image via SHA-384, driving iterative chaos to produce dual sequences for processing the 3D color image matrix through dimensionality reduction, row-wise closed-loop scrambling-diffusion, reconstruction, chaos-guided $\infty$-shaped transformation, isomorphic substitution, and column-wise closed-loop processing. Simulation results confirm the algorithm's effectiveness in protecting color images, with security analysis demonstrating robust resistance against chosen-plaintext, brute-force, and differential attacks.

**Table 12. Running time of different images.**

| Image | size | Encryption time (s) | Decryption time (s) |
|---|---|---|---|
| House | 256×256 | 0.5384 | 0.5398 |
| Baboon | 512×512 | 1.3035 | 1.3274 |
| Peppers | 512×512 | 1.2914 | 1.2950 |
| San Diego | 1024×1024 | 4.2535 | 4.3169 |

Future work will focus on optimizing encryption efficiency and extending the proposed algorithm to multi-image, medical image, and video encryption scenarios for secure communication applications.

## Acknowledgments

We would like to extend our heartfelt appreciation to the editorial board and anonymous reviewers for their meticulous evaluation, valuable insights, and constructive recommendations, which have significantly enhanced the quality of this work.

## Author contributions

**Conceptualization:** Xiaoqiang Zhang.

**Data curation:** Feng Zhao, Xiaoqiang Zhang.

**Formal analysis:** Xiaoqiang Zhang.

**Funding acquisition:** Feng Zhao, Fang Zhu.

**Methodology:** Feng Zhao.

**Project administration:** Fang Zhu.

**Software:** Feng Zhao.

**Writing – original draft:** Feng Zhao.

**Writing – review & editing:** Feng Zhao, Fang Zhu.

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
