## [Decision Letter · Decision Letter 0]

1 Jan 2026

Dear Dr. Zhang,

Thank you for submitting your manuscript to PLOS ONE. After careful consideration, we feel that it has merit but does not fully meet PLOS ONE’s publication criteria as it currently stands. Therefore, we invite you to submit a revised version of the manuscript that addresses the points raised during the review process.

We look forward to receiving your revised manuscript.

Kind regards,

Haris Calgan, Ph.D.

Academic Editor

PLOS One

Journal Requirements:

“The work is supported partly by the Key Research Project of Natural Science in Universities of Anhui Province under Grant No. 2023AH051807, Grant No.2022AH051864, and Grant No. 2024AH050611, and Anhui Xinhua University Level Research Project under Grant No.2022zr003.”

4. We note that your Data Availability Statement is currently as follows: “All relevant data are within the manuscript and its Supporting Information files.”

5. We notice that your supplementary figures are uploaded with the file type 'Figure'. Please amend the file type to 'Supporting Information'. Please ensure that each Supporting Information file has a legend listed in the manuscript after the references list.

6. Please amend either the abstract on the online submission form (via Edit Submission) or the abstract in the manuscript so that they are identical.

7. We note you have included a table to which you do not refer in the text of your manuscript. Please ensure that you refer to Table 1 in your text; if accepted, production will need this reference to link the reader to the Table.

Additional Editor Comments:

Based on the advice received, we feel that your manuscript could be reconsidered for publication should you be prepared to incorporate major revisions. When preparing your revised manuscript, you are asked to carefully consider the reviewer comments which are attached, and submit a list of responses to the comments.

Reviewer's Responses to Questions

**Comments to the Author**

1. Is the manuscript technically sound, and do the data support the conclusions?

Reviewer #1: Yes

Reviewer #2: Yes

Reviewer #3: Yes

2. Has the statistical analysis been performed appropriately and rigorously?

Reviewer #1: Yes

Reviewer #2: Yes

Reviewer #3: Yes

3. Have the authors made all data underlying the findings in their manuscript fully available?

Reviewer #1: Yes

Reviewer #2: Yes

Reviewer #3: Yes

4. Is the manuscript presented in an intelligible fashion and written in standard English?

Reviewer #1: Yes

Reviewer #2: Yes

Reviewer #3: Yes

Reviewer #1: The authors propose a color image encryption scheme using a new 2D chaotic map (2D-CSHS) combining sine, cosine, and exponential terms. The method utilizes an infinity-shaped transformation for pixel scrambling and a closed-loop control model for diffusion. Simulation results on key space, entropy, and robustness are presented to validate the scheme.

Overall, the paper presents a complete scheme, but the novelty of the infinity-shaped scrambling is not well-justified mathematically. The chaotic map has potential stability issues (singularities), and the robustness analysis contradicts standard diffusion theories without sufficient explanation. Therefore, I recommend Major Revision to address the theoretical defects and experimental inconsistencies listed below.

1). In Eq. (3), the term pi-squared divided by x(n-1) is problematic. If the state variable x approaches zero during iteration, the system will overflow or become unstable. You must discuss how to handle this singularity or prove that the trajectory never hits zero in the defined phase space.

2). The proposed 2D-CSHS involves complex transcendental functions (cosine, sine, exponential). Compared to simple Logistic or Tent maps, this computational overhead is high. You should provide a specific floating-point operation count (FLOPs) analysis or comparison to justify using such a heavy map for image encryption.

3). The definition in Eq. (6) Beta = P * Alpha suggests matrix multiplication. If you implement this as actual matrix multiplication, the complexity is O(N^2), which is unacceptable for large images. You must clarify if this is implemented via coordinate mapping O(N) and strictly define the index generation rules.

4). Why is the infinity-shaped path superior to established paths like Hilbert, ZigZag, or Peano curves? The paper only shows the path visually. You need to provide a metric (e.g., degree of scrambling or distance between adjacent pixels) to compare your method against classical scanning curves.

5). In Section 5.8 (Robustness), the decrypted images are clear despite noise/cropping. However, your diffusion is "closed-loop," which typically implies a strong avalanche effect (one bit change affects all). If the algorithm is robust to noise, the diffusion might be weak. Please explain this contradiction theoretically.

6). The key space calculation in Section 5.1 assumes a precision of 10^-15 translates directly to effective key bits. This is a common mistake. In IEEE 754 double-precision standard, the effective entropy is limited (approx 52 bits mantissa). Please revise the calculation based on actual computer representation standards.

7). In Tables 5 and 6 (NPCR/UACI), you only list "passed." This is not scientific. You must list the actual P-values or the specific critical values corresponding to the image size and significance level (alpha=0.05) used in your hypothesis testing to prove the pass. Please refer to Two-dimensional coupling-enhanced cubic hyperchaotic map with exponential parameters: construction, analysis, and application in hierarchical significance-aware multi-image encryption for revisions.

8). While Figure 13 shows flat histograms, visual inspection is insufficient for a high-quality journal. You should calculate and tabulate the Chi-square statistics for the cipher images to quantitatively demonstrate the uniformity of the pixel distribution. Please refer to State-dependent variable fractional-order hyperchaotic dynamics in a coupled quadratic map: a novel system for high-performance image protection for revisions.

9). Table 9 shows encryption takes ~7.5 seconds for a 1024x1024 image. This is quite slow for practical applications (usually milliseconds are required). You should optimize the code or admit this limitation. The heavy reliance on trig functions and matrix reshaping is likely the bottleneck. Please clarify or revise.

10). The text mentions changing "one bit" for NPCR/UACI tests. Did you test changing bits in different positions (MSB vs. LSB)? Chaotic systems can sometimes be less sensitive to LSB changes in floating-point domains. A more comprehensive sensitivity test across different bit planes would be convincing.

11). The direction of diffusion (clockwise/counter-clockwise) is determined by the random number X. Since X is derived from the key/hash, it is deterministic. Does this really add security complexity, or is it just a linear variation? Please discuss the cryptanalytic benefit of this bi-directional feature.

12). In Table 3 and Table 4, you compare your results with references from 2024 and 2025. Ensure that the test images used in those references are identical (same dimensions and source) to yours. If dimensions differ, metrics like NPCR and local entropy cannot be directly compared.

13). A well-crafted abstract should concisely present the research background, identify the research gap, highlight the study’s novel contributions, summarize key findings, particularly critical experimental results, and emphasize the work’s broader significance and value. Please refer to State-dependent variable fractional-order hyperchaotic dynamics in a coupled quadratic map: a novel system for high-performance image protection and Dynamics analysis, synchronization and FPGA implementation of multiscroll Hopfield neural networks with non-polynomial memristor for improvements.

14). The current keyword list does not fully capture the core content and distinctive contributions of the manuscript. The author is encouraged to revise it to better align with the paper’s key themes and technical focus. Please refer to Two-dimensional coupling-enhanced cubic hyperchaotic map with exponential parameters: construction, analysis, and application in hierarchical significance-aware multi-image encryption and Exploiting dynamic vector-level operations and a 2D-enhanced logistic modular map for efficient chaotic image encryption for revisions. For example, the author should consider adding keywords such as "Dynamical analysis " and "security analysis".

15). Given that your work focuses on chaotic encryption, the Introduction should provide a systematic and categorized overview of recent chaotic encryption methods based on new techniques (neural networks, quantum computing, etc.), such as Dynamics analysis, synchronization and FPGA implementation of multiscroll Hopfield neural networks with non-polynomial memristor, State-dependent variable fractional-order hyperchaotic dynamics in a coupled quadratic map: a novel system for high-performance image protection, Two-dimensional coupling-enhanced cubic hyperchaotic map with exponential parameters: construction, analysis, and application in hierarchical significance-aware multi-image encryption, and Exploiting dynamic vector-level operations and a 2D-enhanced logistic modular map for efficient chaotic image encryption. Please refer to the first paragraph of Exploiting robust quadratic polynomial hyperchaotic map and pixel fusion strategy for efficient image encryption for revisions.

16). Cryptanalysis targeting image encryption holds significant reference value for designing encryption schemes. The authors should incorporate recent cryptanalysis studies in the introduction, such as Cryptanalysis of an image encryption cryptosystem based on binary bit planes extraction and multiple chaotic maps, Security analysis of a color image encryption algorithm using a fractional-order chaos, and Cryptanalysis and improvement of the image encryption scheme based on Feistel network and dynamic DNA encoding. Besides, the authors should evaluate whether the proposed work has sufficient practicality and security based on these cryptanalysis works.

17). There are some issues with the presentation and explanation of mathematical symbols and equations. Each equation should be followed by a comma or period, as appropriate. Please refer to State-dependent variable fractional-order hyperchaotic dynamics in a coupled quadratic map: a novel system for high-performance image protection and Dynamics analysis, synchronization and FPGA implementation of multiscroll Hopfield neural networks with non-polynomial memristor for improvements.

18). The conclusion feels underdeveloped and lacks impact. I suggest the authors restructure it, drawing inspiration from Two-dimensional coupling-enhanced cubic hyperchaotic map with exponential parameters: construction, analysis, and application in hierarchical significance-aware multi-image encryption and Exploiting dynamic vector-level operations and a 2D-enhanced logistic modular map for efficient chaotic image encryption, to clearly highlight the study’s core findings, main contributions, existing challenges, proposed solutions, key limitations, and future directions in a more compelling and cohesive way.

19). Please revise and enhance the reference list. Specifically, remove low-quality sources (e.g., papers published in non-SCI journals such as Multimedia Tools and Applications or non-prestigious conferences) and outdated references. Instead, incorporate, introduce, analyze, discuss, and compare the most recent relevant works to strengthen the research background, particularly the discussion of chaotic image encryption and its cryptanalysis studies.

Reviewer #2: This is my comment before publication

a. what is the research gap of this paper?

b. What are the theoretical advantages of ∞-shaped over U-shaped, Z-shaped, or Hilbert-shaped? And is it just a different traversal path, or does it have new cryptographic properties?

c. Discussion image encryption is very low, i suggest author discuss some new work like "https://doi.org/10.1038/s41598-024-80969-z", "https://doi.org/10.1109/ACCESS.2024.3351693" and "https://doi.org/10.1109/ACCESS.2020.3011724"

d. The manuscript over-emphasizes chaotic dynamics analysis without clearly linking them to practical cryptographic strength.

e. Compare cipher-image entropy with and without chaos

f. i suggest author add Cipher-image entropy vs u, r parameters

g. Attacker capability not explained: known-plaintext? chosen-ciphertext?

h. What the attacker knows? What is hidden? and Is SHA-384 enough to prevent known-plaintext attacks?

i. Robustness against noise and cropping is not a standard security requirement for encryption algorithms and may contradict cryptographic diffusion principles.

j. Algorithmic complexity analysis is missing

k. Check typo and grammatical error

Reviewer #3: Paper presents a well-structured color image encryption framework that integrates a newly designed 2D chaotic map with an ∞-shaped transformation and a closed-loop diffusion model. The algorithm is clearly organized, and the encryption–decryption flow is logically consistent. The authors provide extensive experimental evaluation, including entropy, NPCR/UACI, correlation, robustness, and time analysis, which demonstrates careful implementation and validation. The use of plaintext-dependent key generation via SHA-384 strengthens resistance to chosen-plaintext attacks and improves practical security relevance. Overall, the work shows solid effort and technical depth in chaos-based image encryption research.

A few comments are listed below.

1) proposed 2D-CSHS chaotic map is mathematically defined and experimentally analyzed, but its novelty over existing coupled sine–cosine or logistic-based maps is not sufficiently justified. A direct analytical or experimental comparison with closely related 2D hyperchaotic maps is required to clarify what new dynamical property or security advantage this map uniquely provides.

2)bifurcation, Lyapunov exponent, and sample entropy analyses confirm chaotic behavior, but these results are largely descriptive. The paper should explain how the chosen parameter ranges specifically impact encryption strength, rather than only stating that they exhibit chaos. A short sensitivity discussion linking chaotic metrics to cryptographic performance would improve rigor. Next,the ∞-shaped transformation is visually intuitive, but its security role is not formally analyzed. The paper should discuss whether this transformation provides stronger mixing than existing scan-based permutations such as Hilbert, zigzag, or U-shaped scans, especially in terms of inter-channel decorrelation. A quantitative comparison would strengthen this section.

3) The closed-loop control model combines scrambling and diffusion, which is a good design choice. However, the diffusion equations are complex and not easy to verify. A simplified explanation of how error propagation spreads across the loop would help readers understand why this design improves resistance to differential attacks. Also,The key generation process relies heavily on SHA-384 and chaotic iterations, leading to a very large key space. However, the practical independence between hash-derived keys and chaotic parameters is not clearly proven. The authors should clarify whether any correlation exists between these key components under finite precision arithmetic.

4) Finite State Machine based image encryption schemes model the cipher as a finite number of states with deterministic or key-controlled state transitions. While this structure is simple and efficient, the state space is inherently limited. Once the transition function and output mapping are inferred, the system behavior becomes predictable. So, Authos may compare some papers like Enhancing image security via block cyclic construction and DNA based LFSR; Chaos-based medical image encryption scheme using special nonlinear filtering function based LFSR; PwLMμ-TPE: A Visual-Usability and Privacy-Enhanced Thumbnail Preserving Encryption Scheme of Cloud Computing for Consumers. Authors are informed to include a clear security comparison with FSM-based encryption schemes, highlighting how the proposed chaotic and closed-loop diffusion approach avoids finite state predictability and offers stronger resistance to state reconstruction and differential attacks.

5) Insecurity analysis, most metrics are close to ideal values, but comparisons are limited to a small set of reference schemes. Including at least one lightweight or low-complexity encryption scheme as a baseline would make the performance claims more balanced and realistic.The robustness analysis under noise and cropping shows visually acceptable decrypted images, but the security implication of partial information recovery is not discussed. From a cryptographic perspective, successful recovery after severe cropping may also indicate diffusion weakness, which should be critically analyzed.

**Do you want your identity to be public for this peer review?** For information about this choice, including consent withdrawal, please see our For information about this choice, including consent withdrawal, please see our Privacy Policy .

Reviewer #1: No

Reviewer #2: No

Reviewer #3: No

---

## [Author Response · Author response to Decision Letter 1]

23 Feb 2026

Response to Editor’s Comments

We would like to express our sincere thanks for your constructive comments and positive response.

1.Please ensure that your manuscript meets PLOS ONE's style requirements, including those for file naming.

ANS：Yes, we have carefully reviewed and verified it.

2.Please note that PLOS One has specific guidelines on code sharing for submissions in which author-generated code underpins the findings in the manuscript.

ANS: To help readers better understand the algorithm and the article, we have provided the pseudocode of the core algorithm in the manuscript to facilitate program replication. If necessary, you may also contact us for the corresponding code.

“The work is supported partly by the Key Research Project of Natural Science in Universities of Anhui Province under Grant No. 2023AH051807, Grant No.2022AH051864, and Grant No. 2024AH050611, and Anhui Xinhua University Level Research Project under Grant No.2022zr003.”

ANS: We have stated in the cover letter that the funding agency or members of its project team participated in the writing and revision of the paper.

4. We note that your Data Availability Statement is currently as follows: “All relevant data are within the manuscript and its Supporting Information files.”

5. We notice that your supplementary figures are uploaded with the file type 'Figure'. Please amend the file type to 'Supporting Information'. Please ensure that each Supporting Information file has a legend listed in the manuscript after the references list.

ANS：We have revised the file names and re-uploaded them to the Supporting Information, which contains all the graphical results mentioned in the manuscript.

6.Please amend either the abstract on the online submission form (via Edit Submission) or the abstract in the manuscript so that they are identical.

ANS：Thank you for the reminder. I will submit it as required this time.

7.We note you have included a table to which you do not refer in the text of your manuscript. Please ensure that you refer to Table 1 in your text; if accepted, production will need this reference to link the reader to the Table.

ANS: We have carefully reviewed and verified the figures and tables in the manuscript to ensure that each one is cited in the text.

8.If the reviewer comments include a recommendation to cite specific previously published works, please review and evaluate these publications to determine whether they are relevant and should be cited. There is no requirement to cite these works unless the editor has indicated otherwise.

ANS: Thank you for your reminder.

Response to Reviewer 1’s Comments

Firstly, we would like to express our sincere thanks for your constructive comments and positive response. The revised version of the manuscript has been significantly modified based on your comments. The revised parts are marked in red font. We hope you are satisfied with our revisions.

1. In Eq. (3), the term pi-squared divided by x(n-1) is problematic. If the state variable x approaches zero during iteration, the system will overflow or become unstable. You must discuss how to handle this singularity or prove that the trajectory never hits zero in the defined phase space.

ANS: As you noted, according to Eq. (3), xn∈[−1,1] and yn∈[−1,1]. As a result, singularities may occur during the iterative process when xn−1=0. In practical numerical iterations, a regularization method is adopted to address this issue: when |xn-1|< ε, its value is adjusted to  , where ε′ is set to 10−10. This approach effectively avoids divergence caused by singularity. Since only a negligible perturbation is introduced, the overall chaotic dynamics of the

system remain largely unchanged.

2. The proposed 2D-CSHS involves complex transcendental functions (cosine, sine, exponential). Compared to simple Logistic or Tent maps, this computational overhead is high. You should provide a specific floating-point operation count (FLOPs) analysis or comparison to justify using such a heavy map for image encryption.

ANS: To build a more secure and complex image encryption system, this paper designs a novel two-dimensional chaotic system based on the composition of trigonometric and nonlinear functions. Although this system has higher computational complexity compared to classical Logistic or Tent maps, such design is essential to ensure the encryption strength. To quantify its computational demand, we conducted a floating-point operation (FLOPs) analysis on this map. Each iteration requires approximately 12 fundamental floating-point operations (including 4 exponentiations, 4 multiplications, 2 trigonometric functions, and 2 additions). Although the computational overhead is higher than that of one-dimensional maps, a single iteration can still be completed within nanoseconds on modern general-purpose processors, posing no substantial impact on the overall encryption efficiency. More importantly, this system overcomes the inherent limitations of one-dimensional chaotic maps in encryption, such as their small key space and insufficient security. By expanding the parameter space and enhancing the unpredictability of orbits, it significantly improves the security of image encryption while still maintaining real-time performance.

3. The definition in Eq. (6) Beta = P * Alpha suggests matrix multiplication. If you implement this as actual matrix multiplication, the complexity is O(N^2), which is unacceptable for large images. You must clarify if this is implemented via coordinate mapping O(N) and strictly define the index generation rules.

ANS: The definition in Equation (6) is a matrix multiplication. However, there are five types of such transformations in the algorithm, the largest of which involves a matrix of 3 rows and 5 columns. Correspondingly, the transformation matrix P is only of size 15×15, meaning the computational complexity of this matrix operation is on the order of N(152 ). This complexity does not vary with image size; it primarily depends on the maximum transformation matrix used.

4.Why is the ∞-shaped path superior to established paths like Hilbert, ZigZag, or Peano curves? The paper only shows the path visually. You need to provide a metric (e.g., degree of scrambling or distance between adjacent pixels) to compare your method against classical scanning curves.

ANS: The ∞-shaped transform differs from conventional methods such as the Hilbert transform, Zigzag transform, or Peano curve transform in two main aspects. First, it utilizes an inherent three-row matrix transformation to directly achieve inter-channel scrambling among the three color channels. Second, it is divided into five distinct forms based on different column counts, enabling dynamic scrambling effects that effectively enhance the complexity of the scrambling algorithm.

To verify the effectiveness of the ∞-shaped transform compared to classical scrambling methods such as Zigzag, simulation experiments were conducted using a 256×256 house image, with the results presented in Table 3. It presents a comparative analysis of correlation coefficients for both adjacent pixels and inter-channel pixels under different scrambling schemes. It can be observed that the ∞-shaped transform outperforms the Hilbert transform, Zigzag transform, and Peano curve transform in disrupting both spatial correlations among neighboring pixels and statistical dependencies across color channels. Specifically, this scheme substantially reduces the horizontal, vertical, and diagonal correlations of the R, G, and B channels, while also effectively weakening the inter-channel correlations (e.g., R–G, R–B, G–B). These results demonstrate the superior decorrelation capability and scrambling randomness of the ∞-shaped transform. In contrast, the other scrambling methods exhibit limited effectiveness in suppressing inter-channel correlation, with some cases showing negligible impact. Overall, the ∞-shaped transform achieves more comprehensive

performance in image scrambling tasks.

Table 3.  Comparison of correlation coefficients of different scrambling schemes.

Scrambling Path Original

image Hilbert transform Zigzag transform Peano curve transform ∞-shaped transform

Adjacent Pixel Correlation R H 0.9671 0.6984 0.1980 0.7679 0.6638

V 0.9353 0.6975 0.1977 0.6504 0.6696

D 0.9126 0.5613 0.1661 0.5504 0.5659

G H 0.9805 0.7614 0.1669 0.7899 0.7685

V 0.9474 0.7598 0.1674 0.7247 0.7643

D 0.9320 0.6367 0.1478 0.6240 0.5254

B H 0.9820 0.8190 0.2039 0.8404 0.7258

V 0.9749 0.8180 0.2055 0.7881 0.7318

D 0.9625 0.7101 0.2061 0.6903 0.7404

Inter-channel Pixel Correlation R-G 0.6378 0.6378 0.6378 0.6378 0.5420

R-B 0.4823 0.4823 0.4823 0.4823 0.6300

G-B 0.9418 0.9418 0.9418 0.9418 0.6499

5. In Section 5.8 (Robustness), the decrypted images are clear despite noise/cropping. However, your diffusion is “closed-loop”, which typically implies a strong avalanche effect (one bit change affects all). If the algorithm is robust to noise, the diffusion might be weak. Please explain this contradiction theoretically.

ANS: As you have rightly noted, there exists a certain trade-off between robustness and diffusion capability. Generally, if the diffusion avalanche effect is strong, robustness tends to be weaker, whereas a weaker avalanche effect often corresponds to better robustness. However, based on our research group’s analysis and discussions, we believe that robustness is not solely determined by diffusion but is also closely related to scrambling. In this paper, we propose an image encryption scheme based on an infinite-type transformation diffusion method and a closed-loop control model. Experimental results confirm its excellent encryption performance. In future algorithm research, we will continue to examine the balance between robustness and diffusion avalanche effects, striving to achieve an optimal equilibrium that meets practical application requirements.

6. The key space calculation in Section 5.1 assumes a precision of 10^-15 translates directly to effective key bits. This is a common mistake. In IEEE 754 double-precision standard, the effective entropy is limited (approx 52 bits mantissa). Please revise the calculation based on actual computer representation standards.

ANS: We sincerely thank the reviewer for identifying this critical inaccuracy in our original key space calculation. The reviewer is absolutely correct. Our previous method of equating a decimal precision of 10−15 directly to 1015 possible values was incorrect, as it ignores the finite representation of floating-point numbers in digital computers.

Inspired by your suggestion, we have completely revised the key space analysis in Section 5.1. The new calculation is now based on the IEEE 754 double-precision floating-point standard. We conservatively assume that each floating-point initial value or parameter provides approximately 52 bits of effective entropy (corresponding to the 52-bit mantissa).The key space for the form dependent on floating-point values is now calculated as (252)8=2416.The overall effective key space of the algorithm is taken as this value (2416), which remains far above the 2100 threshold required to resist brute-force attacks.

We have updated the text, removed the incorrect calculation based on 10−15 precision, and cited the relevant standard. We believe the revised analysis is now technically sound and accurately reflects the algorithm’s security against exhaustive key search.

7. In Tables 5 and 6 (NPCR/UACI), you only list “passed”. This is not scientific. You must list the actual P-values or the specific critical values corresponding to the image size and significance level (alpha=0.05) used in your hypothesis testing to prove the pass. Please refer to Two-dimensional coupling-enhanced cubic hyperchaotic map with exponential parameters: construction, analysis, and application in hierarchical significance-aware multi-image encryption for revisions.

ANS: Inspired by your suggestions, after carefully reading the recommended literature, we have thoroughly revised and rewritten the calculation standards and experimental results of the NPCR and UACI metrics in Section 5.7 “Differential Attack Analysis” of the paper. The revised content has been highlighted in red within the text.

8. While Figure 13 shows flat histograms, visual inspection is insufficient for a high-quality journal. You should calculate and tabulate the Chi-square statistics for the cipher images to quantitatively demonstrate the uniformity of the pixel distribution. Please refer to State-dependent variable fractional-order hyperchaotic dynamics in a coupled quadratic map: a novel system for high-performance image protection for revisions.

ANS: Inspired by your suggestions,We have supplemented the content of Section 5.4 “Histogram Analysis” by adding experiments on the chi-square test of pixel values in the encrypted images. The calculation results show that all encrypted images passed the test, which provides more objective evidence of the encryption algorithm’s capability to resist statistical analysis attacks.

9. Table 9 shows encryption takes ~7.5 seconds for a 1024x1024 image. This is quite slow for practical applications (usually milliseconds are required). You should optimize the code or admit this limitation. The heavy reliance on trig functions and matrix reshaping is likely the bottleneck. Please clarify or revise.

ANS: Inspired by your valuable suggestions, we have optimized the encryption algorithm accordingly and retested it under the same experimental conditions. Currently, the encryption time for a 1024×1024 image has been reduced to approximately 4.2 seconds, representing a significant improvement over the original version. We fully acknowledge that encryption efficiency in practical applications typically needs to reach the millisecond level, and there remains room for further optimization in current performance. The speed of image encryption is influenced by multiple factors, including algorithm implementation efficiency, hardware conditions, and image size characteristics. In our future work, we will continue to focus on optimizing both the algorithm and the code to further enhance the real-time performance and practicality of the encryption process.

10. The text mentions changing “one bit” for NPCR/UACI tests. Did you test changing bits in different positions (MSB vs. LSB)? Chaotic systems can sometimes be less sensitive to LSB changes in floating-point domains. A more comprehensive sensitivity test across different bit planes would be convincing.

ANS: In the NPCR/UACI tests against differential attacks, we have systematically evaluated the sensitivity of the algorithm to changes in different bit positions. Specifically, modifications were tested for the Least Significant Bit (LSB, i.e., bit 0), the middle bit (bit 3), and the Most Significant Bit (MSB, i.e., bit 7). The experimental results show that the encryption system exhibits good sensitivity across all tested bit planes, with no observed insensitivity to LSB changes in floating-point arithmetic environments. This confirms that the proposed algorithm possesses comprehensive and reliable differential sensitivity.

11. The direction of diffusion (clockwise/counter-clockwise) is determined by the random number X. Since X is derived from the key/hash, it is deterministic. Does this really add security complexity, or is it just a linear variation? Please discuss the cryptanalytic benefit of this bi-directional feature.

ANS: Indeed, in our algorithm, the diffusion direction (clockwise/counterclockwise) is determined by the random variable X. X is generated through iterative computation from the input initial value and the hash value of the plain image, producing a chaotic sequence. Clockwise and

---

## [Decision Letter · Decision Letter 1]

6 Mar 2026

Color Image Encryption Algorithm Based on ∞-Shaped Transformation and Closed-Loop Control model

PONE-D-25-63738R1

Dear Dr. Zhang,

We’re pleased to inform you that your manuscript has been judged scientifically suitable for publication and will be formally accepted for publication once it meets all outstanding technical requirements.

Kind regards,

Haris Calgan, Ph.D.

Academic Editor

PLOS One

Additional Editor Comments (optional):

Reviewers' comments:

Reviewer's Responses to Questions

**Comments to the Author**

Reviewer #1: All comments have been addressed

Reviewer #2: All comments have been addressed

Reviewer #3: All comments have been addressed

2. Is the manuscript technically sound, and do the data support the conclusions?

Reviewer #1: Yes

Reviewer #2: Yes

Reviewer #3: Yes

3. Has the statistical analysis been performed appropriately and rigorously?

Reviewer #1: Yes

Reviewer #2: Yes

Reviewer #3: Yes

4. Have the authors made all data underlying the findings in their manuscript fully available?

Reviewer #1: Yes

Reviewer #2: Yes

Reviewer #3: Yes

5. Is the manuscript presented in an intelligible fashion and written in standard English?

Reviewer #1: Yes

Reviewer #2: Yes

Reviewer #3: Yes

Reviewer #1: Dear authors, thank you for your careful and diligent revisions. All my concerns have been addressed. Therefore, I am pleased to recommend the acceptance of your manuscript.

Reviewer #2: The author has carefully addressed the comments and suggestions provided by the reviewer.

Specifically:

- The author has improved the clarity of the problem statement and strengthened the research justification.

- Minor language issues and formatting inconsistencies have been corrected to improve overall readability.

Overall, the revised version demonstrates significant improvement in terms of clarity, structure, and academic rigor. The responses are appropriate, and the manuscript is now substantially stronger compared to the previous version.

Reviewer #3: All the reviewers’ comments have been carefully addressed and incorporated. The manuscript is now suitable for acceptance.

**Do you want your identity to be public for this peer review?** For information about this choice, including consent withdrawal, please see our For information about this choice, including consent withdrawal, please see our Privacy Policy .

Reviewer #1: No

Reviewer #2: No

Reviewer #3: No

---

## [Editor Report · Acceptance letter]

PONE-D-25-63738R1

PLOS One

Dear Dr. Zhang,

I'm pleased to inform you that your manuscript has been deemed suitable for publication in PLOS One. Congratulations! Your manuscript is now being handed over to our production team.

Kind regards,

on behalf of

Assoc. Prof. Dr. Haris Calgan

Academic Editor

PLOS On